# One Less Reason for Filter-Pruning: Gaining Free Adversarial Robustness with Structured Grouped Kernel Pruning

**Shaochen (Henry) Zhong**[1], **Zaichuan You**[*2], **Jiamu Zhang**[*2], **Sebastian Zhao**[*3], **Zachary LeClaire**[2], **Zirui Liu**[1], **Daochen Zha**[1], **Vipin Chaudhary**[2], **Shuai Xu**[2], and **Xia Hu**[1]

[1]Department of Computer Science, Rice University
{shaochen.zhong, zirui.liu, daochen.zha, xia.hu}@rice.edu
[2]Department of Computer and Data Sciences, Case Western Reserve University
{zxy456, jxz1217, zjl16, vipin, sxx214}@case.edu
[3]Electrical Engineering and Computer Sciences, UC Berkeley
sebbyzhao@berkeley.edu

## Abstract

Densely structured pruning methods utilizing simple pruning heuristics can deliver immediate compression and acceleration benefits with acceptable benign performances. However, empirical findings indicate such naïvely pruned networks are extremely fragile under simple adversarial attacks. Naturally, we would be interested in knowing if such a phenomenon also holds for carefully designed modern structured pruning methods. If so, then to what extent is the severity? And what kind of remedies are available? Unfortunately, both questions remain largely unaddressed: no prior art is able to provide a thorough investigation on the adversarial performance of modern structured pruning methods (spoiler: it is not good), yet the few works that attempt to provide mitigation often do so at various extra costs with only to-be-desired performance.

In this work, we answer both questions by fairly and comprehensively investigating the adversarial performance of 10+ popular structured pruning methods. Solution-wise, we take advantage of *Grouped Kernel Pruning (GKP)*'s recent success in pushing densely structured pruning freedom to a more fine-grained level. By mixing up kernel smoothness — a classic robustness-related kernel-level metric — into a modified GKP procedure, we present a one-shot-post-train-weight-dependent GKP method capable of advancing SOTA performance on both the benign and adversarial scale, while requiring no extra (in fact, often less) cost than a standard pruning procedure. Please refer to our GitHub repository for code implementation, tool sharing, and model checkpoints.

## 1 Introduction

Convolutional neural networks (CNNs) have demonstrated solid performance on tasks centered around computer vision. However, with modern CNNs growing in both widths and depths, the issue of over-parameterization has drawn increasing attention due to such networks often requiring large computational resources and memory capacity. To mitigate the burden, network pruning — the study of removing redundant parameters from original networks without significant performance loss — has become a popular approach for its simplicity and directness [LeCun et al., 1989, Blalock et al., 2020, He and Xiao, 2023].

---

* Equal contribution. Order determined alphabetically by last names.

37th Conference on Neural Information Processing Systems (NeurIPS 2023).

Despite the popularity of the pruning field in general, **few prior arts have been available to provide** *improved adversarial robustness* **under the constraint of** *(densely) structured pruning*; even though empirical findings show vanilla structured pruning methods implemented with naïve pruning strategies often experience huge performance drop on such adversarial tasks [Wang et al., 2018, Sehwag et al., 2020, Vemparala et al., 2021]. More concerning, no prior art has made an effort to provide a comprehensive investigation on whether the same phenomenon also exists under carefully designed modern structured pruning methods, where such methods are often capable of delivering excellent benign accuracy retention after pruning (sometimes, even improvements).

Below, we provide a walk-through of why a densely structured method and having an adversarially robust pruned model are preferable and important, to how we developed our solution by leveraging the power of increased structural pruning freedom (grouped kernel pruning) with kernel-level metrics (kernel smoothness). In the later sections of this paper, we replicate and evaluate around 13 popular densely structured pruning methods and variants against various white box (evasion) adversarial attacks, where our proposed method showcases clear dominance.

## 1.1 Structured v.s. Unstructured Pruning: Accuracy-Efficiency Trade-off

Most of the existing CNN pruning methods can be roughly categorized into *structured* and *unstructured* pruning. Note we said *roughly* because there is no universally agreed delineation between structured and unstructured pruning methods. The general consensus is that methods considered more unstructured often enjoy a higher degree of freedom on where to apply their pruning strategies (e.g., weight-level pruning) and therefore result in better accuracy retention.

In contrast, structured pruning methods often prune weights in a grouped manner following some kind of architecturally defined constraints (e.g., filter-level pruning). Compared to their unstructured counterparts, structured pruning methods are more hardware-friendly and easier to obtain compression and acceleration on commodity hardware, though at the cost of worse accuracy retention. This difference is due to unstructurally pruned networks' tendency to have pruned (zeroed) parameters randomly distributed in the weight matrix, leading to poor data reuse and locality [Yang et al., 2018]. Such unstructurally pruned networks struggle to have wall-clock time speed up without supports like custom-indexing, special operation design, sparse operation libraries, or even dedicated hardware setups [Yang et al., 2018, Han et al., 2016, He and Xiao, 2023].

Among all structured pruning methods, one popular line of research is to produce pruned networks that are entirely dense, a.k.a. ***densely structured, where the pruned weights are stored in the normal dense tensor format***. Such format of a pruned network is considered to be most library/hardware-friendly and, therefore, most deployable in a practical context. With such significant benefits, densely structured pruning methods consist of the absolute majority of structured pruning methods.

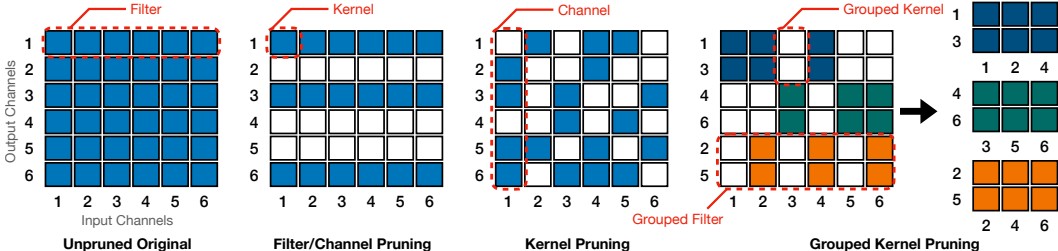

Figure 1: Visualization of different pruning granularities.

Densely structured pruning methods come with different pruning granularities, where a significant portion of prior arts prune at a filter or channel level. These two types of pruning are historically considered to be the limit of densely structured pruning, as showcased in Figure 1: if we go down one more level, we will have kernel pruning; however, its pruned network is not dense. This is until recently, authors from Zhong et al. [2022] utilized kernel pruning with grouped convolution, where they prune at a grouped kernel level to ensure an entirely dense pruned network. To the best of our knowledge, grouped kernel pruning carries the highest degree of pruning freedom among all densely structured pruning methods.

## 1.2 Pruned Models Are Fragile Under Adversarial Attacks — But Why Do We Care?

Empirical findings like Wang et al. [2018] suggest that although pruned neural networks may have acceptable benign accuracy, they are often more vulnerable to adversarial attacks. Adversarial robustness is recognized as a long-standing metric to evaluate model quality, as a model with undesired adversarial robustness can be easily exploited to produce wrong and potentially harmful output, resulting in fairness and accountability issues.

**We would argue such robustness properties are especially valued under the context of (structured) pruning**, where pruned models are often deployed to resource-constraint devices with less central oversight available and requiring execution in a more real-time manner. Imagine if an OCR model for real-time check redeeming can be maliciously exploited to read the number 1 as 9; the result will surely be unpleasant for many parties involved.

To alleviate such a problem (though not under a pruning context), prior arts like Wang et al. [2020b] demonstrate the adversarial robustness of a convolutional network is largely correlated to its sensitiveness to high-frequency components (HFC), where such sensitiveness can be mitigated with some simple kernel-level operations like kernel smoothness. More on this in Section 2.

## 1.3 Solution: Grouped Kernel Pruning with Adversarial-Robustness-Boosting Kernel Metrics

With the recent *Grouped Kernel Pruning (GKP)* framework pushing the pruning freedom of densely structured pruning to a (close) kernel-level [Zhong et al., 2022], we explore the unique possibility of mixing up adversarial-robustness-boosting kernel metrics — such as kernel smoothness — into the procedure of GKP. We present Smoothly Robust Grouped Kernel Pruning (SR-GKP), a densely structured pruning method that works in a simple post-train one-shot manner, but is often capable of delivering competitive benign performance and much stronger adversarial performance against SOTA filter and channel pruning methods that require more sophisticated procedures. Solution-wise, our main claims and contributions are:

- **Free improvement on adversarial robustness.** Our method has no extra (in fact, often less) cost compared to a standard pruning method, making the gained adversarial robustness entirely free.
- **One-shot & post-train & weight-dependent: the simplest procedure with most compatibility.** Our method is a one-shot-post-train-weight-dependent[1] structured pruning method for CNN that follows the classic train - prune - fine-tune procedure, where all excessive components are pruned all at once before fine-tuning. This means it is compatible with any trained CNNs (as it does not interfere with the training pipeline) yet straightforward to execute, as the pruning procedure before fine-tuning does not rely on access to data but only the trained model's parameters.
- **Raise attention to the important but overlooked field of adversarially robust structured pruning.** Our method is among the few structured pruning methods capable of delivering pruned networks with improved adversarial performance — a field with severe problems, but receives little recognition or solutions.

On the investigation side, we are the first to comprehensively reveal:

- **Drastic adversarial performance difference under a similar benign report**. We found that while different carefully designed modern densely structured pruning methods may showcase similar benign performance, most are done so at the cost of adversarial robustness.
- **One less reason for filter/channel pruning: further endorsing GKP**. Filter and channel pruning methods have dominated the field of densely structured pruning for years; our work — together with Zhong et al. [2022] and Park et al. [2023] — showcased that when done right, grouped kernel pruning-based methods are superior under both benign and adversarial tasks, making it a promising direction for future densely structured pruning research.

---

[1]Note the term **"one-shot"** can be ambiguous under a pruning context. Some literature, including ours, refer to "one-shot" as the pruning procedure is done all at once before fine-tuning [Liu et al., 2019, Frankle and Carbin, 2019, Renda et al., 2020]; yet, some other work refers to "one-shot" as without requiring a fine-tuning stage [Chen et al., 2023, Frantar et al., 2023]. We hereby clarify that our proposed method still requires a fine-tuning stage after pruning. Further, Only-Train-Once (OTO) [Chen et al., 2023] — a popular series of iterative from-scratch pruning methods known for its automatically applicable property — utilizes the term "one-shot" in a manner in line with "one-stop," highlighting its end-to-end nature and model-agnostic capability. For disambiguation, we hereby clarify that our method is, in fact, limited to CNN pruning, requires a trained model to start with, and does not possess such model-agnostic or automatically applicable properties.

For added bonuses, we are the first ones to reproduce and comprehensively report the benign and adversarial performances of multiple structured pruning methods under a fair setting. We believe the lack of such fair and comprehensive reports (on both benign and adversarial tasks) is mainly due to the lack of user-friendly tools. Thus, alongside our method implementation and checkpoint files, we also provide the pruning community **a lightweight open-sourced tool capable of a plug-and-play style of testing different victim models with various adversarial attacks while supporting all procedures a modern pruning method may require**.

## 2 Related Work and Discussion

Due to the page limitation, we will discuss related work regarding white box evasion adversarial attacks, adversarially robust structured pruning, and grouped kernel pruning. Other related topics, such as the compression/acceleration implications of structured and unstructured pruning methods, input component frequency with kernel smoothness, as well as other related ML efficiency and AI safety literature, will be introduced in Appendix B.

**Adversarial Attacks.** Neural networks are known to be vulnerable to adversarial attacks, i.e., a small perturbation applied to the inputs can mislead models to make wrong predictions Szegedy et al. [2014], Goodfellow et al. [2014]. In practice, popular adversarial attacks can often be categorized as white-box and black-box evasion attacks. The difference being white-box attacks have access to the entirety of the model, including input features, architectures, and model parameters, while black-box attacks' access is often constrained (e.g., only input-output pair). Thus, white-box attacks are almost always more effective and efficient than black-box ones; in fact, many classic black-box attacks are constructed in a way to approximate the information that is directly accessible by white-box attacks (e.g., gradient) [Chen et al., 2020]. We opt for white-box attacks for the scope of this paper, given one significant use case of structurally pruned models is edge-device deployments, where the model is more likely to be accessed. Also, white-box attacks generally offer a more straightforward workflow with harder challenges posed.

**Structured Pruning for Adversarial Robustness.** Structured pruning methods, which arguably carry the most practical significance, have been heavily studied throughout the years [Molchanov et al., 2017, Yu et al., 2018, He et al., 2019, Wang et al., 2019a,b, Lin et al., 2019, He et al., 2018, Li et al., 2021, Zhong et al., 2022]. Despite their popularity, few of them focus on adversarial robustness. To the best of our knowledge, there are only four prior arts presenting structured pruning methods while claiming improved performance on adversarial robustness metrics [Vemparala et al., 2021, Ye et al., 2019, Sehwag et al., 2020, Zhao and Wressnegger, 2023]. Unfortunately, Vemparala et al. [2021] does not have a public repository for code, Ye et al. [2019] does not have any experiment on standard `BasicBlock` ResNets for comparative investigation despite their popularity. [Blalock et al., 2020], Sehwag et al. [2020] and Zhao and Wressnegger [2023] mostly propose unstructured methods with only a few structurally pruned ablation studies conducted on limited model-dataset combinations, with some of their structured pruning implementation not published.

The lack of traffic, infrastructure, or baseline in this area has undoubtedly created deterrents to all interested scholars. To fill the gap, **we provide the community an open-sourced toolkit capable of testing various victim models against different adversarial attacks under a pruning context, coming with 10+ popular structured pruning methods already integrated for comparative evaluation. Model checkpoints are also available for direct evaluations.**

**Grouped Kernel Pruning.** Our work relies on the GKP framework — particularly inspired by the recent work of Zhong et al. [2022]. The core of GKP is grouped kernel pruning and reconstruction to a grouped convolution format, where this combination has been explored a few times under the context of structure pruning.

Specifically, Zhong et al. [2022] proposed their take on grouped kernel pruning with a three-stage procedure: filter clustering, generating then deciding which grouped kernel pruning strategy to employ, then reconstructing to grouped convolution format via permutation. This particular framework solved the previous drawback of requiring a complex procedure, yet a rich set of experiment results was showcased to demonstrate its performance advantages against many SOTA structured pruning methods. Our proposed method is largely enabled by the extra pruning freedom that GKP provides.

Outside of Zhong et al. [2022], concurrent work like Zhang et al. [2022a] and follow-up work like Park et al. [2023] have showcased the excellent benign performance of different GKP implementations under a benign context.

# 3 Proposed Method

## 3.1 Motivation: Pruning May Amplify Overfitting to High-Frequency Components

Wang et al. [2020b] suggests CNNs are prone to overfitting high-frequency components (HFC) of inputs — a type of feature that is not robust yet can be easily replicated with adversarial perturbations. Though Wang's finding is towards an unpruned model, such phenomenon can be found, and in fact, even amplified, under a structural pruning setting.

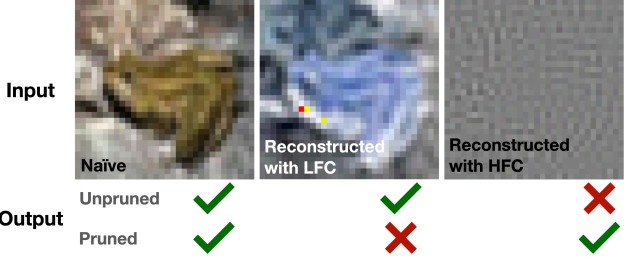

Figure 2: A `frog` figure from CIFAR-10 test set in its original, LFC, and HFC-reconstructed formats. The output indicates the correctness of classification results when testing through an unpruned and a L1Norm pruned ResNet-56 (pruning rate ≈ 43.75% to be consistent with Table 2).

A classic demonstration of such phenomena can be seen in Figure 2. Despite the `frog`-labeled figure reconstructed with only low-frequency components showing visible resemblance to its original benign format, a ResNet-56 model pruned by L1Norm filter pruning [Li et al., 2017] cannot classify it correctly. However, such pruned models can somehow correctly classify the same input reconstructed with only HFCs, even if it already lost all semantics of a frog to a human audience. This indicates a model pruned by methods without having adversarial robustness in consideration is more likely to overfit to HFC.

Table 1: Unpruned and structurally pruned ResNet-56 v. HFC/LFC-reconstructed CIFAR-10 test set. $\theta$ represents the cutoff threshold (for $0 \leq \theta \leq 1$). With a higher $\theta$, the HFC-reconstructed images will exclusively include more high-frequency information; vice-versa for a lower $\theta$. Pruning rate ≈ 43.75%; pruned model checkpoints are taken from Table 2.

| INPUT | UNPRUNED BASELINE | CC PRUNED | NPPM PRUNED | L1NORM-B PRUNED |
|---|---|---|---|---|
| FULL ($\theta = 0.0$) | 93.24 | **94.04** | 93.55 | 92.62 |
| HFC ($\theta = 0.3$) | 77.05 | **80.22** | 78.08 | 79.83 |
| HFC ($\theta = 0.5$) | 50.77 | **57.49** | 55.47 | 56.06 |
| HFC ($\theta = 0.7$) | 22.79 | 25.78 | 21.92 | **27.15** |

We emphasize that the above example (Figure 2) is not a cherry-picked one. As shown in Table 1, by reconstructing the entire test set of CIFAR-10 with solely their high-frequency components, we find that a structurally pruned model is more prone to fitting HFCs than its unpruned counterpart under various settings. This is potentially because HFCs are easily learnable features under a benign setting, so pruned models want to "make most use" of their remaining weights given the reduced network capacity, and therefore become even more overfitted to HFCs by treating them as short-cut features.

**Kernel smoothness as an indicator for learning from HFC.** Fortunately, Wang et al. [2020b] suggests *kernel smoothness* is highly correlated to the learning of HFC. Specifically, Wang et al. [2020b] found that a CNN with "smoother" kernels — where neighbor weights within a 2D kernel have less of a value difference — will reduce the overfitting of HFCs, thus making the model more robust against adversarial perturbations. For the ease of the following conversation, we define the kernel smoothness of a convolution kernel $k$ for $k \in \mathbb{R}^{H \times W}$ to be:

$$\text{smoothness}(k) = \sum_{i=1}^{h \times w} \sum_{\substack{j \in \text{values} \\ \text{border with } k_i}} \left| k_j^2 - k_i^2 \right|, \tag{1}$$

where $H$ and $W$ are the kernel dimensions; in most CNNs, such dimensions are set to $3 \times 3$.

The finding of kernel smoothness and adversarial robustness presents a unique opportunity under the context of GKP. Prior to GKP, densely structured pruning is almost always done at a filter/channel level, where kernel-level metrics/operation have little bearing when relaxed (see Appendix C.1.3). However, GKP prunes at a (close) kernel level, where a kernel-level metric may still retain its power.

## 3.2 Mixing Smoothness into Grouped Kernel Pruning Procedure

It is natural to want to encapsulate some kernel-level operations/metrics — in this case, kernel smoothness — into the procedure of GKP. However, the challenge comes with how we can do it in an efficient and effective manner. Particularly, how can we achieve improved adversarial performance without sacrificing benign tasks?

For the ease of illustration, let $\boldsymbol{W}^\ell \in \mathbb{R}^{C_{\text{out}}^\ell \times C_{\text{in}}^\ell \times H^\ell \times W^\ell}$ be the weight of the $\ell$-th convolutional layer, which consist of $C_{\text{out}}^\ell$ filters, with each filter consisting of $C_{\text{in}}^\ell$ number of $H^\ell \times W^\ell$ 2D kernels. According to Zhong et al. [2022], a standard GKP procedure has two potential stages:

Stage 1: **Filter grouping stage**: where the $C_{\text{out}}^\ell$ filters are clustered into $n$ equal-sized filter groups $\{\boldsymbol{FG}_i^\ell, \boldsymbol{FG}_j^\ell, \dots, \boldsymbol{FG}_n^\ell\}$, with each filter group $\boldsymbol{FG}^\ell \in \mathbb{R}^{C_{\text{out}}^\ell/n \times C_{\text{in}}^\ell \times H^\ell \times W^\ell}$.

Stage 2: **Pruning strategies obtaining/pruning stage**: where the pruning method generates a set of "candidate" grouped kernel pruning strategies to be evaluated and select from; a grouped kernel is defined as a $\boldsymbol{GK} \in \boldsymbol{FG}^\ell$ with $\boldsymbol{GK}$ having a shape of $C_{\text{out}}^\ell/n \times 1 \times H^\ell \times W^\ell$. Finally, we evaluate all collected candidate strategies and decide which to pursue.

However, how to apply kernel smoothness-related criteria to such stages is a non-trivial question. Our empirical results from some proof-of-concept experiments suggest some naïve applications either will not work at all or will only work by significantly sacrificing the performance on benign tasks: e.g., if we simply replace Stage 2 above by pruning the grouped kernels with greater $\sum_{k \in \boldsymbol{GK}} \text{smoothness}(k)$, we will experience a huge drop on benign accuracy (Appendix C). Therefore, we must drive our attention to discover some more sophisticated ways of mixing up such criteria into the above stages.

### 3.2.1 Smoothness Snaking: Is Filter Clustering the Only Right Answer for GKP?

Per TMI-GKP [Zhong et al., 2022], filters within the same convolutional layer are grouped into several equal-sized groups by a *filter clustering schema*, which consist of some different combinations of dimensionality reduction and clustering techniques. The motivation of grouping by clustering is natural as it establishes a preferable search space for pruning algorithms, where the power of pruned components can likely be retrained via similar unpruned components within the same group. However, later procedures of TMI-GKP (e.g., Stage 2 per Section 3.2) then seek to maintain a diverse representation of kept components within each filter group, **which begs the question: is filter clustering the only right answer for GKP? If finding a diverse set of unpruned grouped kernels is the goal, why not have filter groups with filter diversity to start with?** And if the answer to the first question is "No." How can we utilize this opportunity to mix up with kernel smoothness criteria?

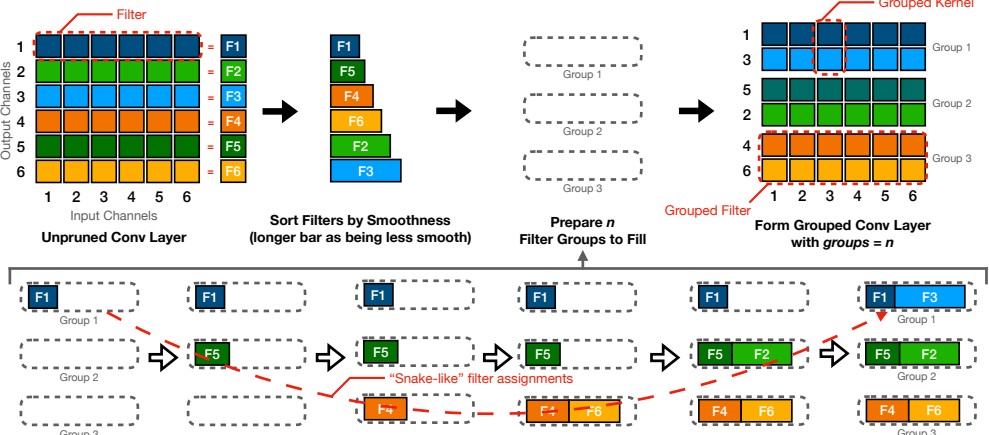

Figure 3: Smoothness Snaking For Filter Grouping (Stage 1)

Grouped-based pruning will always seek out some kind of balance among groups to avoid a skewed distribution where all important components are distributed to certain groups [Zhong et al., 2022]. Following this principle, we want our filter groups to have balanced smoothness — so that we do not end up having any filter group that is "over" or "under-smoothed" to start with for pruning. However, this is equivalent to the *partition problem* [Korf, 1998], which is known to be NP-hard. Given the filter grouping stage is often robust to adjustments[2], we proposed to sort filters according to their smoothness, then assign them iteratively in an S-shaped "snaking" manner across a predefined number of filter groups, as shown in Figure 3, namely ***Smoothness Snaking***. Empirical results suggest although smoothness snaking may not be as optimal as the dynamic clustering scheme in Zhong et al. [2022] in terms of benign performance, it is able to provide better adversarial robustness under adversarial attacks and is much faster to execute (Table 8) due to the absence of dimensionality reduction & clustering procedures (Appendix C). We consider this to be a successful mix-up.

### 3.2.2 Smooth Beam Greedy GKP Search

Knowing that the pruning strategies obtaining/decision stage (Stage 2 per Section 3.2) of GKP is sensitive to tampering, a direct application of smoothness criteria would not work (Appendix C). This indicates the distance-based *cost* formula (Equation 5) proposed in Zhong et al. [2022] carries a significant influence on the performance of a pruned network, which is unsurprising given the vast popularity of distance-based pruning arts. Since we can't directly replace this *cost* formula, we propose to widen the capacity of pruning strategies obtaining stage with smoothness in mind while keeping the decision stage akin to the *cost* formula. By doing this, the chances of more "smoothness-aware" pruning strategies being employed are increased.

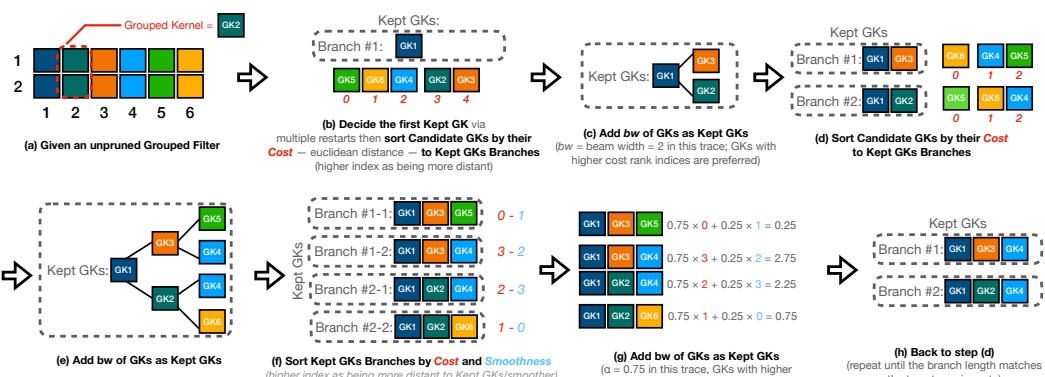

Figure 4: Smoothness-aware Beam Greedy Search for Grouped Kernel Pruning (Stage 2).

Again, to keep our approach simple, we implemented a custom beam search element to consider more grouped kernels during each advancement (Figure 4). Per each iteration, all gathered candidate grouped kernel pruning strategies will be evaluated against a mixture of smoothness and cost criteria, where only a $Beam_{\text{width}}$ amount of grouped kernels will be kept for further advancement (until the desired pruning ratio is reached). The scoring formula to determine which subset of strategies may be kept in the beam is defined as (larger scores are better/preferred):

$$\underset{\text{mix-up}}{\text{Score}}(\boldsymbol{GK}_{\text{branch}}, \alpha) = \alpha \cdot \phi\big(\text{Cost}(\boldsymbol{GK}_{\text{branch}})\big) + (1 - \alpha) \cdot \phi\big(\text{Smoothness}(\boldsymbol{GK}_{\text{branch}})\big), \quad (2)$$

where $\phi\big(\texttt{Criterion}(\boldsymbol{GK}_{\text{branch}})\big)$ for $\texttt{Criterion} \in \{\text{Cost}, \text{Smoothness}\}$ represents the rank of such $\boldsymbol{GK}_{\text{branch}}$ when all collected $\boldsymbol{GK}_{\text{branch}}$ candidates are sorted according to the given $\texttt{Criterion}$ in a descending order. So $\phi\big(\text{Smoothness}(\boldsymbol{GK}_{\text{branch}})\big) = 0$ would suggest this particular $\boldsymbol{GK}_{\text{branch}}$ has a greater smoothness ranking (a.k.a. "less smooth") then all other $\boldsymbol{GK}_{\text{branch}}$ candidates in considerations. $\alpha$ is a tunable balancing parameter that adjusts the importance of one criterion over

---

[2]In Zhong et al. [2022], various filter grouping strategies — including random assignment — were proposed. Many of them tend to perform reasonably well, albeit less performant nor stable than the proposed TMI-driven solution.

the other. The Smoothness($\boldsymbol{GK}_{\text{branch}}$) equation is simply the sum of all kernels within all $\boldsymbol{GK}_{\text{kept}}$ over Equation 1; the Cost($\boldsymbol{GK}_{\text{branch}}$) equation is a modified version of Equation 5 in Zhong et al. [2022], where we removed some hyperparameters for simplicity and to reduce tuning workload. Please refer to Figure 4 and Appendix C.3 for more details.

## 4 Experiments and Results

### 4.1 Experiment Setups

We evaluate the efficacy of our method on ResNet-32/56/110 with the `BasicBlock` implementation, ResNet-50/101 with the `BottleNeck` implementation, and VGG-16 [He et al., 2016, Simonyan and Zisserman, 2015]. For datasets, we choose CIFAR-10 [Krizhevsky, 2009], Tiny-ImageNet [Wu et al., 2017], and ImageNet-1k [Deng et al., 2009] for a wide range of coverage. For all compared methods and under most model-dataset combinations, we tried our best to replicate them with a $\approx 300$ epochs (except for ImageNet, where we only employ 100 epochs) of fine-tuning/retraining budget while maintaining all other settings either identical or proportional to their original publications.

### 4.2 Compared Methods and Evaluating Criteria

We evaluate our proposed method against up to 13 popular densely structured pruning methods and variants shown in Appendix D. Whenever possible, we produce pruned-and-fine-tuned (or retrained) models upon **identical unpruned baseline models** with similar post-prune MACs and Params. Then, we compare their inference accuracy on benign inputs as well as adversarially-perturbed inputs powered by *FGSM* and *PGD* attacks in various perturbation budgets and intensities [Goodfellow et al., 2014, Madry et al., 2018].

One reporting mechanism that is probably unique to our paper — in comparison to standard pruning art under the benign space — is **for some of our methods, results of multiple epochs are reported, each showcasing a method's peak performance against different evaluation metrics (a.k.a. "superscore").** This is because for benign tasks, following Li et al. [2017] and He et al. [2019], only the epoch checkpoint with the best benign accuracy needs to be reported. But under an adversarial context, if a pruning method is capable of producing more than one fully pruned model during the fine-tuning/retraining stage, oftentimes, the best performer per each evaluation metric does not overlap. We believe it is responsible to report them all as there are no dominant evaluation metrics in our experiments; however, this is an important advantage for methods that can generate multiple usable pruned models (e.g., one-shot pruning) over methods that can only generate a few or just one fully pruned model (e.g., layer-wise iterative pruning). We survey the availability of such checkpoints and their epoch cutoff in Table 10: "Fully Pruned Epoch" column.

We also comprehensively investigate each pruning method against a checklist of questions as presented in Table 10 of Appendix D.1.3, including pruning granularity, procedure, when is the first fully pruned epoch, zero-masked or hard-pruning, and many other important questions. **This investigation, along with the epoch budget constraint and baseline control, should provide our audience with a more leveled playing field for fair and informed methods comparison.**

Further, to facilitate digesting mass results reported across many methods/variants and against various metrics, we present our large-scale empirical evaluation in three different ways. Take `BasicBlock` ResNets on CIFAR-10 as an example (with pruning rate $\approx 43.75\%$), we present the raw results as Table 11, 12, and 13; where we visualize them accordingly as Figure 6, 7, and 8. Last, we make a rank chart, Table 3, to provide a straightforward gauge of the competitiveness among different pruning methods.

### 4.3 Results and Analysis

Our abbreviated results — Table 2 and Table 4 — showcased the performance of various modern SOTA methods as well as our proposed methods on the two most popular model-dataset combinations [Blalock et al., 2020]. For ResNet-56 on CIFAR-10, our method outperformed every other method on all evaluating criteria, except for PGD; as RAP-ADMM [Ye et al., 2019] outperformed SR-GKP significantly with $61.16\%$ v. $44.85\%$. However, it is worth noting that RAP-ADMM does adversarial training on PGD-perturbed data, so it is not surprising that it performs well against the seen type of attack. Unfortunately, it seems like the adversarially trained RAP-ADMM cannot generalize its

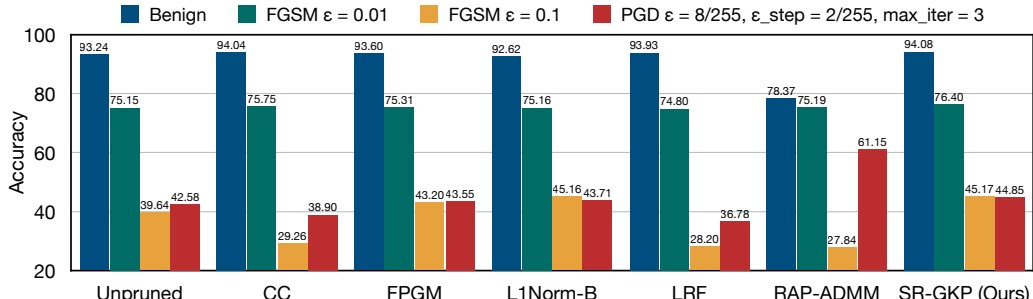

Figure 5: Visualization of ResNet-56 on CIFAR-10 with pruning rate ≈ 43.75% — note this plot is done in a "superscore" manner for a concise presentation; the four bars of a method may not belong to the same model checkpoint.

Table 2: ResNet-56 on CIFAR-10. All pruning methods are performed on the same baseline model. "Best (a)" represents the performance of a model checkpoint that meats the showcased MACs/Params reduction that performs the best against criterion (a).

| Method | Criterion | Benign | (a) $FGSM_{\varepsilon=0.01}$ | (b) $FGSM_{\varepsilon=0.1}$ | (c) $PGD_{max\_iter=3}^{\varepsilon=8/255,\ \varepsilon_{step}=2/255}$ | MACs (M) | Params (M) |
|---|---|---|---|---|---|---|---|
| Unpruned | - | 93.24 | 75.15 | 39.64 | 42.58 | 126.561 | 0.853 |
| CC [Li et al., 2021] | Best Benign | 94.04 | 74.78 | 29.25 | 37.85 | 69.837 | 0.616 |
| | Best (a) | 93.73 | 75.75 | 29.26 | 38.86 | | |
| | Best (b) | 93.70 | 75.56 | 29.89 | 38.90 | | |
| FPGM [He et al., 2019] | Best Benign | 93.60 | 75.31 | 43.20 | 43.55 | 71.661 | 0.482 |
| | Best (a) | 93.37 | 76.28 | 44.96 | 44.66 | | |
| | Best (b) | | | | | | |
| HRank [Lin et al., 2020] | Best Benign | 92.27 | 72.32 | 19.11 | 32.51 | 79.237 | 0.584 |
| | Best (b) | | | | | | |
| | Best (a) | 92.07 | 72.59 | 18.94 | 32.21 | | |
| L1Norm-B [Li et al., 2017] | Best Benign | 92.62 | 72.97 | 41.30 | 41.79 | 72.115 | 0.586 |
| | Best (a) | 91.94 | 75.16 | 42.49 | 43.71 | | |
| | Best (b) | 91.70 | 74.40 | 45.16 | 43.41 | | |
| LRF [Joo et al., 2021] | Best Benign | 93.93 | 73.47 | 25.59 | 34.86 | 71.009 | 0.490 |
| | Best (a) | 93.68 | 74.80 | 27.39 | 36.78 | | |
| | Best (b) | 93.63 | 74.06 | 28.20 | 35.98 | | |
| NPPM [Gao et al., 2021] | Best Benign | 93.55 | 74.82 | 29.07 | 37.12 | 70.843 | 0.601 |
| | Best (a) | 93.35 | 75.50 | 30.29 | 38.09 | | |
| | Best (b) | 93.43 | 75.27 | 31.18 | 38.13 | | |
| RAP-ADMM[Ye et al., 2019] | - | 78.37 | 75.19 | 27.84 | **61.15** | 71.661 | 0.482 |
| SFP[He et al., 2018] | - | 93.15 | 75.63 | 43.83 | 44.10 | 71.462 | 0.481 |
| TMI-GKP [Zhong et al., 2022] | Best Benign | 93.95 | 75.18 | 42.18 | 43.46 | 71.855 | 0.482 |
| | Best (a) | 93.37 | 75.88 | 42.55 | 43.74 | | |
| | Best (b) | 93.66 | 75.74 | 44.09 | 44.51 | | |
| SR-GKP (Ours) | Best Benign | **94.08** | 75.89 | 42.60 | 43.85 | 71.855 | 0.482 |
| | Best (a) | 93.83 | **76.40** | **45.17** | **44.85** | | |
| | Best (b) | | | | | | |

Table 3: Methods ranked against each other on each model with pruning rate ≈ 43.75%, lower is better. "ResNet-XX Mean Rank" means a method's average ranks across four metrics on ResNet-XX; e.g., SR-GKP is ranked #1/#1/#4/#2 for its best performance across benign/FGSM 0.01/FGSM 0.1/PGD metrics on ResNet-56 against other methods, so it'd have a ResNet-56 Mean Rank of (1+1+4+2)/4 = #2. Methods with incomplete presence across the three models are excluded. This table is provided to facilitate the digestion of Table 2, 11, 12, and 13, which consist of raw results.

| Method | ResNet-32 Mean Rank | ResNet-56 Mean Rank | ResNet-110 Mean Rank | All Models Mean Rank |
|---|---|---|---|---|
| CC [Li et al., 2021] | #5.5 | #6.5 | #9.25 | #7.08 |
| DHP [Li et al., 2020] | #10.5 | #8.75 | #12.25 | #10.5 |
| FPGM [He et al., 2019] | #4.75 | #4 | **#2.5** | #3.75 |
| L1Norm-A [Li et al., 2017] | #7.75 | #7 | #8.75 | #7.83 |
| L1Norm-B [Li el al., 2017] | #4 | #6.75 | #8 | #6.25 |
| LRF [Joo et al., 2021] | #7.75 | #8.75 | #7 | #7.83 |
| NPPM [Gao et al., 2021] | #6.5 | #8 | #8.25 | #7.58 |
| OTOv2 (from-scratch) [Chen et al., 2023] | #12.5 | #12.5 | #12.25 | #12.42 |
| OTOv2 (post-train) [Chen et al., 2023] | #6.75 | #8.75 | #6 | #7.17 |
| RAP-ADMM [Ye et al., 2019] | #7 | #7.5 | #7 | #7.17 |
| SFP [He et al., 2018] | #9 | #6.25 | #3 | #6.08 |
| TMI-GKP [Zhong et al., 2022] | #5.25 | #4.25 | #4.25 | #4.58 |
| **SR-GKP (Ours)** | **#3.75** | **#2** | **#2.5** | **#2.75** |

Table 4: ResNet-50 on ImageNet-1k. Note this table includes two baselines: "self-trained" and "`torchvision`". This is because TMI-GKP requires training epoch snapshots, which is not supplied with the `torchvision` pretrained ResNet-50. Methods included in this table are either well-performing methods under the CIFAR-10 experiments or specifically recommended by the reviewers.

| Method | Baseline | Benign | $\text{FGSM}_{\varepsilon=0.001}$ | $\text{FGSM}_{\varepsilon=0.01}$ | $\text{FGSM}_{\varepsilon=0.1}$ | $\text{PGD}_{\text{max\_iter}=3}^{\varepsilon=4/255,\ \varepsilon_{\text{step}}=1/255}$ | MACs (M) | Params (M) |
|---|---|---|---|---|---|---|---|---|
| Unpruned | Self-trained | 75.70 | 67.57 | 25.82 | 16.16 | 6.66 | 4122.828 | 25.557 |
| TMI-GKP [Zhong et al., 2022] | Self-trained | 75.02 | 67.62 | 25.86 | 15.82 | 7.26 | 2725.954 | 17.069 |
| SR-GKP | | **75.29** | **68.02** | **26.45** | **15.83** | **8.08** | 2725.954 | 17.069 |
| Unpruned | `torchvision` | 76.13 | 70.18 | 28.70 | 13.93 | 9.27 | 4122.828 | 25.557 |
| DFPC [Narshana et al., 2022] | `torchvision` | 73.80 | 67.45 | 25.15 | 12.24 | 7.43 | - | - |
| FPGM [He et al., 2019] | `torchvision` | 75.04 | **68.50** | 25.84 | 13.06 | 7.43 | 2641.670 | 18.310 |
| OTOv2 (post-train) [Chen et al., 2023] | `torchvision` | **75.38** | 68.04 | 21.01 | 12.76 | 5.26 | - | - |
| SFP [He et al., 2018] | `torchvision` | 58.50 | 55.72 | 25.82 | 9.01 | **11.00** | 2635.129 | 17.302 |
| SR-GKP | `torchvision` | **75.34** | 68.04 | **26.65** | **15.94** | 7.85 | 2759.672 | 17.803 |

defense to other adversarial attacks, even though they are similar in nature. Also, RAP-ADMM has the worst benign performance across all showcased methods, yet its training time is significantly longer due to the need to perturb its training data on the fly constantly.

Table 2 (and similar experiments showcased in Appendix D) may answer one of our research questions: **are carefully designed modern structured pruning methods also fragile under adversarial attacks? The answer is an unfortunate "Yes,"** as not only do recent structured pruning methods show serious performance drops under adversarial attacks, such drops are often more severe than their predecessors — which often rely on much naïve designs. Figure 5 as well as the ranked chart Table 3 provide a vivid illustration with LRF [Joo et al., 2021] — a 2021 method — showing one of the weakest adversarially robustness across all evaluated methods.

Upon careful comparison, we noticed that SFP [He et al., 2018] and FPGM [He et al., 2019] — two pre-2020 methods — tend to be the best filter pruning methods under the double scrutiny of benign and adversarial tasks. However, this only holds true to smaller scale experiments, as SFP is significantly outperformed by SR-GKP on ResNet-50 on ImageNet for benign accuracy ($58.50\%$ v. $74.34\%$). Though FPGM tends to perform better on the same task, it is still behind SR-GKP in a general sense (Table 4). **We further note ImageNet with adversarial metrics is rarely evaluated under the context of adversarially robust pruning** [Chen et al., 2022, Ye et al., 2019]; this is mainly because many of ImageNet sub-classes are closely related in terms of semantics (e.g., man-eating shark v. tiger shark) [Ozbulak et al., 2021]. We provide the ImageNet results primarily to show SR-GKP can deliver competitive benign ImageNet performance, where most ImageNet results compared in Table 4 are from methods with strong adversarial performance on CIFAR-10.

Last, although TMI-GKP [Zhong et al., 2022] is optimized for benign tasks, and its procedure does not consider adversarial scenarios, it naturally comes with strong adversarial robustness. We believe this has a lot to do with the increased pruning freedom enabled by the GKP granularity, which further highlights the potential of GKP-based methods outside its achievements in the benign space. Due to the page limitation, we hereby only showcase a much-abbreviated version of our experiments. **We strongly encourage our readers to check out our full experiments at Appendix D with a lot more comprehensive coverage on many more dataset-model combinations, as well as ablation studies.**

## 5    Conclusion

Our work studies the area of adversarially robust structured pruning, a topic presenting severe problems but lacking proper recognition or exploration. On the investigation side, we reveal that — just like their naïve predecessors — carefully designed modern structured pruning methods are also fragile under adversarial attacks, yet different pruning methods may yield drastically different adversarial performance while hiding behind similar benign reports. On the solution side, we propose SR-GKP: a simple one-shot GKP method that showcases competitive benign performance with a significant advantage under adversarial attacks against comparable SOTA methods while requiring no extra cost from a pruning procedure perspective.

We believe the overlooked nature of this field is mainly two-fold: the lack of pruning freedom to utilize findings of other fields, and the lack of user-friendly tools and resources for doing adversarial evaluations under a pruning context. We present our take and contribution to both issues by showcasing the capability of GKP-based methods and providing our community with an open-sourced tool and model checkpoints for future studies.

# 6 Acknowledgments

This research was supported, in part, by NSF Awards OAC-2117439, OAC-2112606, IIS-2224843, and IIS-1900990. This work made use of the High Performance Computing Resource in the Core Facility for Advanced Research Computing at Case Western Reserve University. We give our special thanks to the CWRU HPC team for their timely and professional help and maintenance.

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

# A  Limitation and Broader Impact

Note although our investigation revealed serious performance issues under adversarial tasks in terms of structured pruning methods, where our proposed method provides sensible mitigation; our findings are still limited to the benign and artificially perturbed input. Though we expect grouped kernel pruning-based methods to deliver good performance under other reasonable evaluating metrics due to the improved pruning freedom, we will leave a more comprehensive investigation against other robustness metrics for future work. We also provide a brief discussion of such related metrics in Appendix B.4.

# B  Extended Related Work and Discussion

## B.1  Structured v.s. Unstructured Pruning

Pruning methods can be roughly divided into structured pruning and unstructured pruning according to the pruning granularity. Specifically, unstructured pruning often means we prune each weight independently. In contrast, structured pruning bundles weights into groups, then prune the whole group instead of the individual weight (e.g., block-wise Lagunas et al. [2021], channel-wise He et al. [2017], group-wise pruning Zhong et al. [2022]).

Unstructured pruning can maintain the model performance better with the same number of parameters. However, unstructured pruned models yield marginal wall-clock time efficiency or even slower than the unpruned model at the low sparsity regime. This is because unstructured pruned matrices need to be stored in sparse matrix format, as the zeros are randomly distributed in these matrices Yang et al. [2018]. Operations executed on sparse matrices (e.g., sparse matrix multiplication, sparse embedding table look-up) are notoriously inefficient on commodity hardware, e.g., GPUs and CPUs, due to the limited data reuse and random memory access Yang et al. [2018], Han et al. [2016].

In contrast, although structured pruning has less flexibility compared to unstructured one, it is much more hardware/library-friendly since structurally pruned matrices can still be stored in the dense matrix format. Thus, operations executed on structured pruned matrices are the same as those in the unpruned model, which are highly optimized. Consequently, the compression provided by structured pruning can often translate into the real wall-clock time speedup upon proper implementation.

## B.2  Unstructured Pruning for Adversarial Robustness

In Section 2, we introduced the few structured pruning methods which claim increased adversarial robustness, indicating the lack of presence of adversarially robust structured pruning methods. However, many unstructured pruning literature have explored the possibility of pruned models with improved adversarial robustness, such as Sehwag et al. [2020], Zhao and Wressnegger [2023], Ye et al. [2019], Jian et al. [2022], Li et al. [2022], Gui et al. [2019]. Given it is natural for scholars to adopt insights from unstructured pruning methods into a structured context, we hereby provide this list of unstructured pruning arts for inspiration purposes.

## B.3  Learning of High-Frequency Components and its Adversarial Implications

As we consult adversarial-robustness-boosting kernel metrics and operations, we heavily rely on the findings from Wang et al. [2020b], a work that discusses how learning components with different frequencies may affect the adversarial robustness of a CNN, and how some kernel-level metrics like kernel smoothness influence such type of learning. We have elaborated more on this in the Section 3.1 above. Following work can also be referenced when it comes to how component frequency may affect (or be adopted to improve) adversarial robustness, efficiency, and general neural network learning [Wang et al., 2020a, 2023b,a].

## B.4  More on ML efficiency and AI safety

The point of studying adversarially robust structured pruning methods is to progress ML efficiency while being aware of its safety. We note that structured pruning is not the sole method that can achieve efficiency benefits, yet adversarial robustness is nowhere near a comprehensive metric for AI

safety. Approaches such as quantization, network architecture search, knowledge distillation, lossy approximation, efficiency-aware architecture tweak, and transformation to hardware-friendly patterns also provide efficiency gains [Frantar et al., 2023, Lin et al., 2023, Wang et al., 2020a, Zhang et al., 2022b, Liu et al., 2023, Xiao et al., 2023, Dettmers et al., 2023, Dao et al., 2022b,a]; yet aspects like explainability, domain generalization, fairness/bias-mitigation, model/data security, and robustness under a noisy environment are also considered important metrics for building reliable, safe, and trustworthy AI solutions [Wang et al., 2022, Wan et al., 2023, Chuang et al., 2023a,b, Chang et al., 2023c,a, Jiang et al., 2022, Chang et al., 2023b, Tang et al., Zha et al., 2023].

# C  Additional Details on Proposed Method

## C.1  Naïve Mix-Up Attempts of Kernel Smoothness

As per Section 3.2, there are generally two stages in a GKP procedure: *Filter Grouping* and *Grouped Kernel Pruning*. The following experiments shall attempt to mix-up kernel smoothness criteria into each of such stages, and we can therefore find out which stage is "friendly" to such mix-up operations and how such operations should look like.

### C.1.1  During GKP Filter Grouping (Stage 1)

In TMI-GKP [Zhong et al., 2022], the filter grouping stage is driven by the *tickets magnitude increase* score, known as *TMI-driven Clustering*. We utilize it as a baseline to investigate whether our smoothness-aware filter grouping operation — *Smoothness Snaking* (Section 3.2.1 and Figure 3) — can maintain the baseline performance and provide improvements. Note, we denote TMI-GKP's grouped kernel pruning scheme (GK Pruning) as *Greedy* in the following Table 5.

Table 5: Comparison of different *Filter Grouping* methods in GKP. All compared models are pruned to identical MACs/Params for fairness.

| Model | Filter Grouping | GK Pruning | Criterion | Benign | (a) FGSM$_{\varepsilon=0.01}$ | (b) FGSM$_{\varepsilon=0.1}$ |
|---|---|---|---|---|---|---|
| | Unpruned | | - | 92.80 | 71.93 | 31.35 |
| ResNet-32 | TMI-driven Clustering | Greedy | Best Benign | **92.99** | 69.15 | 19.59 |
| | | | Best (a) | 92.03 | 70.61 | 15.85 |
| | | | Best (b) | 92.77 | 69.90 | 30.51 |
| | Smoothness Snaking | Greedy | Best Benign | 92.77 | **71.47** | **30.64** |
| | | | Best (a) | | | |
| | | | Best (b) | 92.65 | **70.82** | **30.65** |
| | Unpruned | | - | 93.24 | 75.15 | 39.64 |
| ResNet-56 | TMI-driven Clustering | Greedy | Best Benign | **93.95** | 75.18 | 42.18 |
| | | | Best (a) | 93.37 | 75.88 | 42.55 |
| | | | Best (b) | 93.66 | 75.74 | 44.09 |
| | Smoothness Snaking | Greedy | Best Benign | 93.62 | 75.68 | 41.21 |
| | | | Best (a) | 93.40 | **76.05** | 43.03 |
| | | | Best (b) | 93.44 | 75.69 | **44.62** |

It can be observed that though Smoothness Snaking may yield a slightly lower benign performance than its TMI-driven baseline, it may improve the adversarial robustness when used with the same grouped kernel pruning procedure. We would also note Smoothness Snaking is significantly faster (up to 1,600x) than TMI-driven clustering due to the absence of dimensionality reduction and clustering procedure (see Table 8 for details).

### C.1.2  During GKP Pruning Strategies Obtaining/Pruning (Stage 2)

Following the section above, here we investigate the performance of different *Grouped Kernel Pruning* methods. Our proposed method is denoted as *Smooth Beam Greedy* (see Figure 4 for details), TMI-GKP's grouped kernel pruning method is denoted as *Greedy* as above, and *Least Smooth* represents a vanilla adaptation of smoothness-driven pruning, where the grouped kernels that are least smooth are pruned.

Table 6: Comparison of different *Grouped Kernel Pruning* methods in GKP. All compared models are pruned to identical MACs/Params for fairness.

| Model | Filter Grouping | GK Pruning | Criterion | Benign | (a) FGSM$_{\varepsilon=0.01}$ | (b) FGSM$_{\varepsilon=0.1}$ |
|---|---|---|---|---|---|---|
| ResNet-32 | Unpruned | | - | 92.80 | 71.93 | 31.35 |
| | Smoothness Snaking | Greedy | Best Benign | 92.77 | 71.47 | 30.64 |
| | | | Best (a) | | | |
| | | | Best (b) | 92.65 | 70.82 | 30.65 |
| | Smoothness Snaking | Least Smooth | Best Benign | 90.09 | 63.91 | 17.22 |
| | | | Best (a) | 88.90 | 70.41 | 15.33 |
| | | | Best (b) | 48.37 | 47.80 | 28.55 |
| | Smoothness Snaking | Smooth Beam Greedy | Best Benign | **92.97** | 70.57 | 29.31 |
| | | | Best (a) | 92.88 | **71.52** | 30.39 |
| | | | Best (b) | 92.86 | 70.79 | **31.32** |
| ResNet-56 | Unpruned | | - | 93.24 | 75.15 | 39.64 |
| | Smoothness Snaking | Greedy | Best Benign | 93.62 | 75.68 | 41.21 |
| | | | Best (a) | 93.40 | 76.05 | 43.03 |
| | | | Best (b) | 93.44 | 75.69 | 44.62 |
| | Smoothness Snaking | Least Smooth | Best Benign | 90.97 | 73.32 | 24.95 |
| | | | Best (a) | 90.01 | 74.63 | 14.10 |
| | | | Best (b) | 90.59 | 73.60 | 26.40 |
| | Smoothness Snaking | Smooth Beam Greedy | Best Benign | **94.08** | 75.89 | 42.60 |
| | | | Best (a) | **93.83** | **76.40** | **45.17** |
| | | | Best (b) | | | |

From Table 6, we may tell that *Smooth Beam Greedy* may significantly improve the adversarial robustness of the pruned networks, yet, it also provides remedies to the decrease in benign performance due to the smoothness snaking operation. It may also be worth noting that the *Least Smooth* operation is extremely detrimental to almost all tracking metrics, **suggesting a vanilla mix-up is inappropriate.**

### C.1.3   Bonus Investigation: Is Smoothness-aware Filter Pruning Possible?

One main purpose of our paper is to endorse the potential of GKP under adversarial tasks, after [Zhong et al., 2022, Zhang et al., 2022a, Park et al., 2023] showcased the power of GKP in a benign context, thus "one less reason for filter pruning." But to make such a claim proper, we will need to investigate whether it is possible to do the same smoothness-aware mix-up with a filter pruning procedure.

Here in Table 7, we use SFP [He et al., 2018] as the baseline, which is considered one of the strongest filter pruning methods on CIFAR-10 [Krizhevsky, 2009]. Then, we try to apply the same ranked-based kernel-smoothness mix-up algorithm as in Equation 2 to find out if such a strong filter pruning baseline can withstand the same mix-up under a filter level.

Table 7: Filter pruning method SFP [He et al., 2018] applied with the same mix-up algorithm in Equation 2. Note "CSB" represents "cost smoothness balancer", which is also $\alpha$ in Equation 2 — so a higher CSB means more biased towards the distance-based *Cost* metrics. All compared models are pruned to identical MACs/Params for fairness.

| Model | Method | Benign | FGSM$_{\varepsilon=0.01}$ | FGSM$_{\varepsilon=0.1}$ | PGD$_{\text{max\_iter}=3}^{\varepsilon=8/255,\ \varepsilon_{\text{step}}=2/255}$ |
|---|---|---|---|---|---|
| ResNet-32 | SFP | 91.94 | 69.25 | 30.34 | 30.18 |
| | CSB = 0.5 | 83.49 | 40.86 | 21.12 | 24.28 |
| | CSB = 0.75 | 91.03 | 65.94 | 22.92 | 25.40 |
| ResNet-56 | SFP | 93.15 | 75.63 | 43.83 | 44.10 |
| | CSB = 0.5 | 81.83 | 60.46 | 19.59 | 28.45 |
| | CSB = 0.75 | 92.26 | 72.11 | 37.42 | 36.64 |
| | CSB = 0.9 | 93.09 | 74.78 | 40.64 | 41.41 |

It can be observed that SFP under such mix-up is completely unusable due to the degradation of benign performance; **which suggests the same mix-up strategy, though effective on GKP, is not transferable to filter pruning.**

## C.2 Speed-Up Analysis

### C.2.1 Pruning Procedure Speed Comparison: TMI-Driven Clustering v.s. Smoothness Snaking

Table 8: Wall-clock runtime comparison between SR-GKP (Ours) and TMI-GKP [Zhong et al., 2022].

| Method | ResNet-32 Group | ResNet-32 Total | ResNet-56 Group | ResNet-56 Total | ResNet-110 Group | ResNet-110 Total |
|--------|-----------------|-----------------|-----------------|-----------------|------------------|------------------|
| TMI-GKP | 47m 6s | 1h 20m 10s | 2h 32m 12s | 2h 36m 22s | 4h 53m 33s | 5h 30m 18s |
| SR-GKP | 3s | 11m 36s | 5s | 20m 43s | 11s | 1h 10m 16s |

In Table 8, we showcased the significant runtime advantage of SR-GKP to TMI-GKP due to the absence of clustering and dimensionality reduction procedures.

### C.2.2 Inference Speed Discussion: Standard Convolution v.s. Grouped Convolution

In the realm of structured pruning, there are two types of popular implementations in practice, which may result in different readings on efficiency metrics, such as inference cost reduction (FLOPs/MACs), inference speed-up, and model size compression (number of parameters):

**Masking (soft pruning)**  To deploy a structured binary mask upon the original unpruned model, but clear the gradients of some (or all) pruned components before the weight update step during fine-tuning, therefore theoretically "structural pruning" the model. In this implementation, the pruned model will not reduce in dimension; thus, no acceleration and compression benefits can be directly observed.

This is a popular implementation — as surveyed in Table 10 "Zero-Masked?" column — because some iterative pruning methods would like the pruned/zeroed components to be reactive during fine-tuning, then pruning another set of components instead. If the pruning granularity is structured (e.g., filter-level), the fine-tuned model can be converted to a hard pruned model by the end of the update as defined below.

**Component removal (hard pruning)**  To remove the pruned components entirely, where the pruned model will reduce in size, providing immediate compression benefits.

Our method, SR-GKP, is implemented as hard pruning. This means that if you inspect the weights of a GKP-pruned network, they are in regular shapes, and there is no zeroed weight to be found — a.k.a. "densely structured." Which may lead to a direct reduction in FLOPs/MACs and model size.

In terms of inference acceleration, previous grouped convolution implementations cannot provide speed-up benefits against a standard convolution (even with much fewer MACs/Params) because the standard convolution operator has been extensively optimized, while the grouped one hasn't (e.g., `pytorch` issue #10229 and #18631). But after `torch 2.0`, this is not the case *at large*. For simplicity, here we compare the forward wall-clock between a standard `Conv2d` of (`C_in = 512, C_out = 512, kernel_size = (3, 3)`) with the exact same `Conv2d` but with groups = 2 (meaning half of its kernels are structurally removed).

Table 9: Inference wall-clock comparison between a standard convolution operator and a GKP convolution operator (w/ `groups=2` and pruning rate = 50%). Input size set as (`64, 3, 224, 224`).

| Operator | Forward | Macs | Params |
|----------|---------|------|--------|
| Unpruned Standard Conv | 129.56 ms | 550528 | 359296 |
| GKP-pruned Grouped Conv | 72.96 ms (56.31%) | 275264 (50%) | 179648 (50%) |

As showcased in Table 9, an organic inference time speed-up of grouped convolution over standard convolution can be observed.

(Note, we emphasized *at large* above because we indeed can find shapes that are slower with grouped convolution in `torch 2.0`. We believe this is very much a framework optimization issue that is beyond the scope of our paper and shall be on the road map of the torch community, as well as other more inference-specific frameworks, granted their already observed improvements.)

### C.3 SR-GKP Procedure

### C.3.1 Simplifying the Cost Formula from TMI-GKP

For better readability, we hereby follow the notation of TMI-GKP [Zhong et al., 2022] (Equation 4), where we assume $V^*$ represents a set of kept grouped kernels provided by a pruning strategy, where $\boldsymbol{g}^\ell$ the convolutional layer in question. The quality $V^*$, under our design, is deemed by:

$$\underset{\text{grouped kernel pruning}}{\text{Score}}(V^*, \boldsymbol{g}^\ell) = \sum_{s_u, s_v \in \binom{V^*}{2}} w(s_u, s_v) - \beta \Big( \sum_{s_i=1 \in V^*} w(p_i, s_i) \Big). \tag{3}$$

Where $w(s_u, s_v)$ represents the Euclidean distance between grouped kernels $s_u$ and $s_v$, but $s_i$ represents the kept grouped kernel that has the least $w(p_i, s_i)$ to a pruned grouped kernel $p_i$. Thus, the former term of this equation calculates the inner distance sum of grouped kernels within strategy $V^*$, and the latter term represents the outer distance between a pruned grouped kernel and its closest kept grouped kernel. Intuitively, we would like the former term to be large, as we would prefer our ideal $V^*$ to have great diversity. Following the same idea, we would like to have the latter term small, as we want the kept kernels to cover the representation power of a certain pruned kernel. By using a $-\beta$ to connect two term, we have $V^*_{\text{best}} = \arg\max_{V^*}(\text{Score}(V^*, \boldsymbol{g}^\ell))$ for all $V^*$s obtained in Stage 2 (Figure 4).

Though similar, we differ from TMI-GKP [Zhong et al., 2022] (Equation 4) in two spots: first, we set $\beta = \binom{V^*}{2}/p_{\text{num}}$, where $p_{\text{num}}$ represent the number of pruned grouped kernels in layer $\boldsymbol{g}^\ell$, balancing the two terms automatically. Secondly, for the latter term of Equation 3, we only match one pruned kernel $p_i$ with one kept kernel $s_i$, instead of several of them. We made these modifications for the main purpose of removing hyperparameters. We kept the grouped kernel strategy selection stage similar to TMI-GKP, but only expanded its search scope using the Smooth Beam Greedy search (Figure 4), because prior investigation suggests this stage is sensitive to smoothness-aware mix-up operations.

# D Extended Experiments and Results

## D.1 Preliminary

### D.1.1 Details of Experiment Setups

For all experiments on VGG-16 and ResNet-32/56/110, we aim to provide all pruning methods an around 300 epochs fine-tuning/retraining budget. Experiments conducted on ResNet-50/101 are budgeted with 100 epochs. We allow compared methods to utilize their own training schedule and vanilla SGD optimizer setup.

SR-GKP utilizes an initial `lr` of 0.01, with a step size of 100 or 30, depending on if it is on the 300 epoch or 100 epoch schedule. `BasicBlock` ResNets with VGG use a weight decay of `5e-4` but `BottleNeck` ResNets use a weight decay of `1e-4`. SR-GKP strictly employs a batch size of 64 for all CIFAR-10 experiments, 128 for Tiny-ImageNet experiments, and 256 for ImageNet-1k experiments.

### D.1.2 Details of Evaluation Criteria

We manually ensure all pruned models have a similar MACs/Params reduction from the identical baseline models. Then, following Li et al. [2017], we report all model checkpoints — yielded during the pruning procedure — that may reach the target MACs/Params reduction. For performance (accuracy) evaluation, we test each model against FGSM [Goodfellow et al., 2014] and PGD [Madry et al., 2018] under various settings and intensities.

### D.1.3 Details of Compared Methods

We provide a method overview in Table 10 to provide our readers with a more comprehensive understanding of such methods.

Table 10: Details of Compared Methods. Note "C/F/GK" under "Type" represents Channel/Filter/Grouped Kernel Pruning. Whether a method requires a "Special Setup" is determined by whether it follows the most vanilla train - prune - fine-tune procedure. "Fully Pruned Epoch" reflects if given a 300 fine-tune/retrain budget, what would be the first epoch that meets the target MACs/Params reduction? (a.k.a. checkpoints after this epoch are reportable as "superscore" readings defined in Section 4.2.) "Zero-Masked?" reflects whether the pruning method can easily yield a compressed model without masking (a.k.a. hard pruning).

| Method | Venue | Type | Procedure | Special Setup? | Zero-Masked? | Fully Pruned Epoch |
|---|---|---|---|---|---|---|
| CC [Li et al., 2021] | CVPR | C | One-shot | Y(requires data) | N | 1 |
| DHP [Li et al., 2020] | ECCV | F | Iterative (from-scratch) | Y (hypernet) | Y | 100 |
| FPGM [He et al., 2019] | CVPR | F | Iterative | N | Y | 1 |
| GAL [Lin et al., 2019] | CVPR | F | Iterative | Y (GAN) | Y | close to 300 |
| GReg [Wang et al., 2021] | ICLR | F | One-shot | No | N | 1 |
| HARP [Zhao and Wressnegger, 2023] | ICLR | C | One-shot | Y (adv. training) | N | 1 |
| HRank [Lin et al., 2020] | CVPR | F | Iterative | N | Y | 325 or 327 |
| L1Norm [Li et al., 2017] | ICLR | F | One-shot | Y (dynamic pruning rate) | N | 1 |
| LRF [Joo et al., 2021] | AAAI | C | One-shot | Y (requires data, adding 1x1, dark knowledge) | N | 1 |
| NPPM [Gao et al., 2021] | CVPR | C | One-shot | Y (hypernet) | N | 1 |
| OTOv2 [Chen et al., 2023] | ICLR | F | Iterative (from-scratch) | N | N | 300 |
| RAP-ADMM [Ye et al., 2019] | ICCV | F | Iterative | Y (adv. training) | Y | 151 |
| SFP [He et al., 2018] | IJCAI | F | Iterative | Y (soft pruning) | Y | 300 |
| TMI-GKP [Zhong et al., 2022] | ICLR | GK | One-shot | N | N | 1 |
| SR-GKP (Ours, 2023) | NeurIPS | GK | One-shot | N | N | 1 |

Further, following the background introduction of pruning implementation in Appendix C.2.2, we hereby justify why a post-train one-shot hard pruning method — such as SR-GKP — may "require no extra (in fact, often less) cost than a standard pruning procedure."

Under the classic train - prune - fine-tune/retrain paradigm (where our method lives), the training cost is determined at two stages: training the unpruned baseline model and fine-tuning the pruned model.

SR-GKP carries no additional training cost as it is a post-train pruning method, so it won't affect the training cost of the first stage. As for fine-tuning, the SR-GKP one-shot pruned model is already compressed and will go through a vanilla SGD update with no extra operation; we also specifically ensured that all compared methods are using an identical (or similar to the best of the method's ability) epoch budget for fine-tuning as stipulated in Section 4.2 — thus, it requires no additional (in fact, often less) cost on the fine-tuning stage as well.

For illustration convenience, here we walk through two exemplary comparisons:

Compared to TMI-GKP [Zhong et al., 2022] — the founding method of Grouped Kernel Pruning — a SR-GKP pruned model shares the exact same format with a TMI-GKP pruned model. Both pruned models will go through an identical fine-tuning procedure. In this case, SR-GKP yields no additional training/fine-tuning cost. Additionally, we'd like to direct your attention to Table 8, which indicates SR-GKP has a much faster execution time than TMI-GKP in terms of pruning procedure, thus being a more "executionally efficient" method overall.

Compared to SFP [He et al., 2018] — a popular filter pruning method that exhibits strong adversarial performance — SFP produces a zero-masked model while SR-GKP's pruned model is hard-pruned. Such zeroed weights will reactivate during its retraining/fine-tuning, where the algorithm decides which filter to zero out between epochs. In this case, the one-shot pruned SR-GKP induces significantly less cost as a) it does not require weight updates to the full model, but only on a compressed one; and b) it does not perform extra pruning operations during the fine-tuning/retraining stage.

## D.2 Additional Experiments

### D.2.1 ResNet-32/56/110 on CIFAR-10 with pruning rate $\approx 43.75\%$ (in addition to Table 2)

Please refer to Table 11 with Figure 6, Table 12 with Figure 7, and Table 13 with Figure 8 for details. We also provide Table 3 in a ranked chart format for an easy digest of the three aforementioned tables.

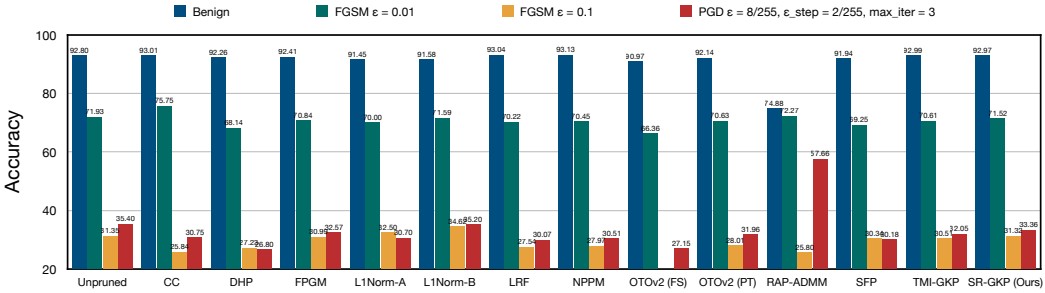

Figure 6: "Superscore" visualization of ResNet-32 on CIFAR-10 with pruning rate $\approx 43.75\%$

Table 11: Full experiments of ResNet-32 on CIFAR-10 with pruning rate $\approx 43.75\%$. All pruning methods are performed on the same baseline model. "Best (a)" represents the performance of a model checkpoint that meets the showcased MACs/Params reduction and performs best against criterion (a).

| Method | Criterion | Benign | (a) $FGSM_{\varepsilon=0.01}$ | (b) $FGSM_{\varepsilon=0.1}$ | (c) $PGD_{max\_iter=3}^{\varepsilon=8/255, \varepsilon_{step}=2/255}$ | MACs (M) | Params (M) |
|---|---|---|---|---|---|---|---|
| Unpruned | - | 92.80 | 71.93 | 31.35 | 35.40 | 69.479 | 0.464 |
| CC [Li et al., 2021] | Best Benign | 93.01 | 70.06 | 25.26 | 30.40 | 39.261 | 0.312 |
| | Best (a) | 92.69 | 70.92 | 25.84 | 30.75 | | |
| | Best (b) | 92.82 | 70.50 | 26.79 | 31.20 | | |
| DHP [Li et al., 2020] | Best Benign | 92.26 | 67.25 | 21.35 | 26.12 | 40.091 | 0.283 |
| | Best (a) | 91.88 | 68.14 | 26.64 | 26.80 | | |
| | Best (b) | 91.80 | 67.62 | 27.23 | 26.65 | | |
| FPGM [He et al., 2019] | Best Benign | 92.41 | 69.75 | 29.61 | **32.32** | 39.352 | 0.262 |
| | Best (a) | 92.06 | 70.84 | 29.74 | **32.57** | | |
| | Best (b) | 92.23 | 69.96 | 30.99 | **32.43** | | |
| GAL [Lin et al., 2019] | - | 90.31 | 72.21 | 31.68 | 38.32 | 38.80 | 0.233 |
| L1Norm-A [Li et al., 2017] | Best Benign | 91.45 | 68.04 | 24.03 | 28.71 | 39.861 | 0.252 |
| | Best (a) | 91.03 | 70.00 | 25.92 | 30.70 | | |
| | Best (b) | 90.38 | 67.61 | **32.50** | 29.67 | | |
| L1Norm-B [Li et al., 2017] | Best Benign | 91.58 | 67.04 | 25.24 | 28.91 | 39.630 | 0.313 |
| | Best (a) | 91.03 | **71.59** | 29.11 | 35.20 | | |
| | Best (b) | 90.59 | 69.67 | **34.62** | 34.02 | | |
| LRF [Joo et al., 2021] | Best Benign | **93.04** | 69.38 | 25.82 | 30.07 | 38.791 | 0.260 |
| | Best (a) | 92.63 | 70.22 | 25.13 | 29.74 | | |
| | Best (b) | 92.79 | 69.21 | 27.54 | 28.64 | | |
| NPPM [Gao et al., 2021] | Best Benign | **93.13** | 69.88 | 26.03 | 29.75 | 39.605 | 0.326 |
| | Best (a) | 92.88 | 70.45 | 27.27 | 30.41 | | |
| | Best (b) | 92.89 | 70.08 | 27.97 | 30.51 | | |
| RAP-ADMM [Ye et al., 2019] | - | 74.88 | 72.27 | 25.80 | **57.66** | 39.370 | 0.271 |
| OTOv2 (from scratch) [Chen et al., 2023] | - | 90.97 | 66.36 | 17.28 | 27.15 | - | - |
| OTOv2 (post train) [Chen et al., 2023] | - | 92.14 | 70.63 | 28.01 | 31.96 | - | - |
| SFP [He et al., 2018] | - | 91.94 | 69.25 | 30.34 | 30.18 | 40.375 | 0.266 |
| TMI-GKP [Zhong et al., 2022] | Best Benign | 92.99 | 69.15 | 19.59 | 28.09 | 39.545 | 0.263 |
| | Best (a) | 92.03 | 70.61 | 15.85 | 29.44 | | |
| | Best (b) | 92.77 | 69.90 | 30.51 | **32.05** | | |
| SR-GKP (Ours) | Best Benign | **92.97** | 70.57 | 29.31 | 32.01 | 39.545 | 0.263 |
| | Best (a) | 92.88 | **71.52** | 30.39 | **33.36** | | |
| | Best (b) | 92.86 | 70.79 | **31.32** | **32.89** | | |

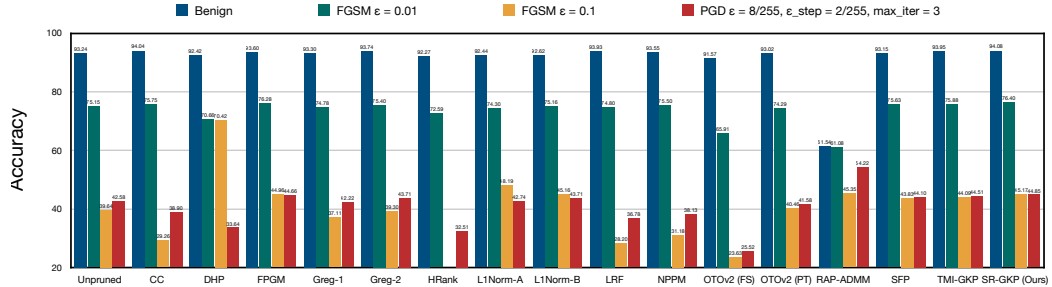

Figure 7: "Superscore" visualization of ResNet-56 on CIFAR-10 with pruning rate ≈ 43.75%

Table 12: Full experiments of ResNet-56 on CIFAR-10 with pruning rate ≈ 43.75%. All pruning methods are performed on the same baseline model. "Best (a)" represents the performance of a model checkpoint that meets the showcased MACs/Params reduction and performs best against criterion (a).

| Method | Criterion | Benign | (a) FGSM$_{\varepsilon=0.01}$ | (b) FGSM$_{\varepsilon=0.1}$ | (c) PGD$_{\text{max\_iter}=3}^{\varepsilon=8/255,\ \varepsilon_{step}=2/255}$ | MACs (M) | Params (M) |
|---|---|---|---|---|---|---|---|
| Unpruned | - | 93.24 | 75.15 | 39.64 | 42.58 | 126.561 | 0.853 |
| CC [Li et al., 2021] | Best Benign | 94.04 | 74.78 | 29.25 | 37.85 | 69.837 | 0.616 |
| | Best (a) | 93.73 | 75.75 | 29.26 | 38.86 | | |
| | Best (b) | 93.70 | 75.56 | 29.89 | 38.90 | | |
| DHP [Li et al., 2020] | Best Benign | 92.42 | 69.50 | 29.93 | 32.56 | 73.289 | 0.480 |
| | Best (a) | 92.34 | 70.66 | 30.17 | 33.64 | | |
| | Best (b) | 92.07 | 31.54 | 70.42 | 32.83 | | |
| FPGM [He et al., 2019] | Best Benign | 93.60 | 75.31 | 43.20 | 43.55 | 71.661 | 0.482 |
| | Best (a) | 93.37 | 76.28 | 44.96 | 44.66 | | |
| | Best (b) | | | | | | |
| GAL [Lin et al., 2019] | - | 91.27 | 76.38 | **47.32** | **47.36** | 98.24 | 0.700 |
| HRank [Lin et al., 2020] | Best Benign | 92.27 | 72.32 | 19.11 | 32.51 | 79.237 | 0.584 |
| | Best (b) | | | | | | |
| | Best (a) | 92.07 | 72.59 | 18.94 | 32.21 | | |
| L1Norm-A [Li et al., 2017] | Best Benign | 92.44 | 73.00 | 41.00 | 40.76 | 67.995 | 0.487 |
| | Best (a) | 91.65 | 74.30 | 43.59 | 42.74 | | |
| | Best (b) | 91.34 | 71.94 | 48.19 | 42.10 | | |
| L1Norm-B [Li et al., 2017] | Best Benign | 92.62 | 72.97 | 41.30 | 41.79 | 72.115 | 0.586 |
| | Best (a) | 91.94 | 75.16 | 42.49 | 43.71 | | |
| | Best (b) | 91.70 | 74.40 | 45.16 | 43.41 | | |
| LRF [Joo et al., 2021] | Best Benign | 93.93 | 73.47 | 25.59 | 34.86 | 71.009 | 0.490 |
| | Best (a) | 93.68 | 74.80 | 27.39 | 36.78 | | |
| | Best (b) | 93.63 | 74.06 | 28.20 | 35.98 | | |
| NPPM [Gao et al., 2021] | Best Benign | 93.55 | 74.82 | 29.07 | 37.12 | 70.843 | 0.601 |
| | Best (a) | 93.35 | 75.50 | 30.29 | 38.09 | | |
| | Best (b) | 93.43 | 75.27 | 31.18 | 38.13 | | |
| RAP-ADMM [Ye et al., 2019] | - | 78.37 | 75.19 | 27.84 | **61.15** | 71.661 | 0.482 |
| OTOv2 (from scratch) [Chen et al., 2023] | - | 91.57 | 65.91 | 23.63 | 25.52 | 79.780 | 0.480 |
| OTOv2 (post train) [Chen et al., 2023] | - | 93.02 | 74.29 | 40.46 | 41.58 | 66.196 | 0.554 |
| SFP [He et al., 2018] | - | 93.15 | 75.63 | 43.83 | 44.10 | 71.462 | 0.481 |
| TMI-GKP [Zhong et al., 2022] | Best Benign | 93.95 | 75.18 | 42.18 | 43.46 | 71.855 | 0.482 |
| | Best (a) | 93.37 | 75.88 | 42.55 | 43.74 | | |
| | Best (b) | 93.66 | 75.74 | 44.09 | 44.51 | | |
| SR-GKP (Ours) | Best Benign | **94.08** | 75.89 | 42.60 | 43.85 | 71.855 | 0.482 |
| | Best (a) | 93.83 | **76.40** | **45.17** | 44.85 | | |
| | Best (b) | | | | | | |

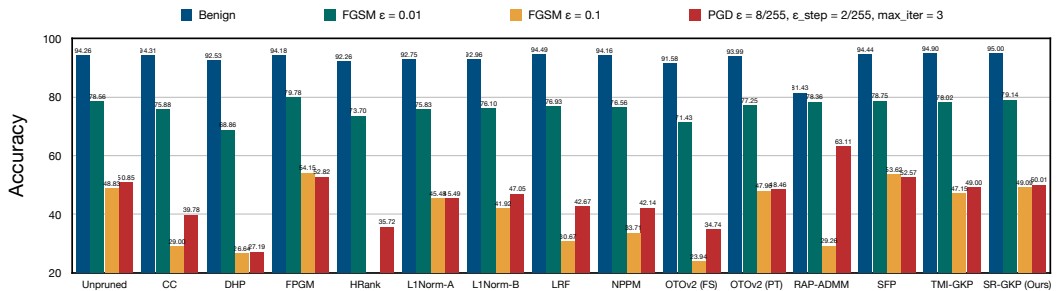

Figure 8: "Superscore" visualization of ResNet-110 on CIFAR-10 with pruning rate ≈ 43.75%

Table 13: Full experiments of ResNet-110 on CIFAR-10 with pruning rate ≈ 43.75%. All pruning methods are performed on the same baseline model. "Best (a)" represents the performance of a model checkpoint that meets the showcased MACs/Params reduction and performs best against criterion (a).

| Method | Criterion | Benign | (a) FGSM$_{\varepsilon=0.01}$ | (b) FGSM$_{\varepsilon=0.1}$ | (c) PGD$_{max\_iter=3}^{\varepsilon=8/255,\ \varepsilon_{step}=2/255}$ | MACs (M) | Params (M) |
|---|---|---|---|---|---|---|---|
| Unpruned | - | 94.26 | 78.56 | 48.83 | 50.85 | 254.995 | 1.728 |
| CC [Li et al., 2021] | Best Benign | 94.31 | 75.12 | 27.31 | 38.24 | 144.414 | 1.046 |
| | Best (a) | 94.17 | 75.88 | 28.14 | 39.78 | | |
| | Best (b) | 94.23 | 75.52 | 29.00 | 39.64 | | |
| DHP [Li et al., 2020] | Best Benign | 92.53 | 67.94 | 25.08 | 26.83 | 101.350 | 0.612 |
| | Best (a) | 92.21 | 68.86 | 25,51 | 27.19 | | |
| | Best (b) | 92.25 | 67.82 | 26.64 | 26.67 | | |
| FPGM [He et al., 2019] | Best Benign | 94.18 | 79.32 | **52.16** | **52.46** | 114.357 | 0.976 |
| | Best (a) | 94.02 | **79.78** | 53.33 | 52.73 | | |
| | Best (b) | 94.10 | 79.70 | **54.15** | 52.82 | | |
| GAL [Lin et al., 2019] | - | 93.42 | 82.34 | 52.79 | **57.55** | 180.677 | 1.186 |
| HRank [Lin et al., 2020] | Best Benign | 92.96 | 73.70 | 16.87 | 35.72 | 158.992 | 1.060 |
| L1Norm-A [Li et al., 2017] | Best Benign | 92.75 | 74.84 | 38.03 | 41.54 | 143.454 | 0.958 |
| | Best (a) | 91.93 | 75.83 | 39.71 | 44.04 | | |
| | Best (b) | 92.21 | 75.26 | 45.48 | 45.49 | | |
| L1Norm-B [Li et al., 2017] | Best Benign | 92.96 | 75.26 | 41.19 | 44.86 | 144.909 | 1.094 |
| | Best (a) | 92.40 | 76.10 | 41.36 | 47.05 | | |
| | Best (b) | 91.71 | 73.58 | 41.92 | 42.69 | | |
| LRF [Joo et al., 2021] | Best Benign | 94.49 | 76.60 | 29.50 | 42.15 | 144.405 | 0.997 |
| | Best (a) | 94.22 | 76.93 | 29.27 | 42.59 | | |
| | Best (b) | 94.38 | 76.71 | 30.67 | 42.67 | | |
| NPPM [Gao et al., 2021] | Best Benign | 94.16 | 76.16 | 32.84 | 41.54 | 146.722 | 1.120 |
| | Best (a) | 94.01 | 76.56 | 33.71 | 42.14 | | |
| | Best (b) | | | | | | |
| RAP-ADMM [Ye et al., 2019] | - | 81.43 | 78.36 | 29.26 | **63.11** | 144.357 | 0.976 |
| OTOv2 (from scratch) [Chen et al., 2023] | - | 91.58 | 71.43 | 23.94 | 34.74 | - | - |
| OTOv2 (post train) [Chen et al., 2023] | - | 93.99 | 77.25 | 47.96 | 48.46 | - | - |
| SFP [He et al., 2018] | - | 94.44 | 78.75 | **53.62** | 52.57 | 144.274 | 0.976 |
| TMI-GKP [Zhong et al., 2022] | Best Benign | **94.90** | 77.50 | 47.15 | 49.00 | 144.551 | 0.976 |
| | Best (a) | **94.63** | 78.02 | 45.95 | 48.81 | | |
| | Best (b) | **94.90** | 77.50 | 47.15 | 49.00 | | |
| SR-GKP (Ours) | Best Benign | **95.00** | 78.01 | 46.53 | 48.86 | 144.551 | 0.976 |
| | Best (a) | 94.69 | **79.14** | 47.49 | **50.01** | | |
| | Best (b) | 94.60 | 78.92 | **49.09** | 49.83 | | |

## D.2.2 ResNet-32/56/110 on CIFAR-10 with pruning rate ≈ 62.5%)

Please refer to Table 15, 16, and 17. We also provide Table 14 in a ranked chart format for an easy digest of the three aforementioned tables.

Table 14: Methods ranked against each other on each model with pruning rate ≈ 62.5%, lower is better. "ResNet-XX Mean Rank" means a method's average ranks across four metrics on ResNet-XX; e.g., SR-GKP is ranked #1/#1/#3/#3 for its best performance across benign/FGSM 0.01/FGSM 0.1/PGD metrics on ResNet-110 against other methods, so it'd have a ResNet-110 Mean Rank of (1+1+3+3)/4 = #2. Methods with incomplete presence across the three models are excluded. This table is provided to facilitate the digestion of Table 15, 16, and 17, which consist of raw results.

| Method | ResNet-32 Mean Rank | ResNet-56 Mean Rank | ResNet-110 Mean Rank | All Models Mean Rank |
|---|---|---|---|---|
| CC [Li et al., 2021] | **#1** | #3.5 | #5.75 | #3.42 |
| DHP [Li et al., 2020] | #3.75 | #7.5 | #7.75 | #6.33 |
| FPGM [He et al., 2019] | #4.25 | #3.75 | **#2** | #3.33 |
| L1Norm-A [Li et al., 2017] | #5.5 | #3.75 | #5.75 | #5.00 |
| L1Norm-B [Li et al., 2017] | #6.25 | #6.25 | #4.25 | #5.58 |
| NPPM [Gao et al., 2021] | #5 | #4 | #5.5 | #4.83 |
| SFP [He et al., 2018] | #7.5 | #5 | #3 | #5.17 |
| **SR-GKP (Ours)** | #2.75 (2nd-best) | **#2.25** | **#2** | **#2.33** |

Table 15: Full experiments of ResNet-32 on CIFAR-10 with pruning rate ≈ 62.5%. All pruning are performed on the same baseline model. "Best (a)" represents the performance of a model checkpoint that meets the showcased MACs/Params reduction and performs best against criterion (a).

| Method | Criterion | Benign | (a) FGSM$_{\varepsilon=0.01}$ | (b) FGSM$_{\varepsilon=0.1}$ | (c) PGD$_{\text{max\_iter}=3}^{\varepsilon=8/255,\ \varepsilon_{\text{step}}=2/255}$ | MACs (M) | Params (M) |
|---|---|---|---|---|---|---|---|
| Unpruned | - | 92.80 | 71.93 | 31.35 | 35.40 | 69.479 | 0.464 |
| CC [Li et al., 2021] | Best Benign | 92.39 | 70.71 | 15.45 | 30.87 | 26.904 | 0.210 |
| | Best (a) | 91.83 | 69.99 | 28.84 | 30.91 | | |
| | Best (b) | 92.01 | 67.97 | 29.07 | 29.29 | | |
| DHP [Li et al., 2020] | Best Benign | 91.73 | 66.87 | 28.00 | 26.71 | - | - |
| | Best (a) | 91.36 | 67.71 | 26.83 | 26.89 | | |
| | Best (b) | 91.31 | 66.84 | 28.75 | 26.39 | | |
| FPGM [He et al., 2019] | Best Benign | 91.32 | 65.41 | 20.91 | 24.97 | - | - |
| | Best (a) | 90.47 | 67.77 | 15.95 | 26.61 | | |
| | Best (b) | 91.04 | 56.51 | 24.47 | 25.68 | | |
| L1Norm-A [Li et al., 2017] | Best Benign | 89.96 | 66.06 | 20.44 | 27.29 | 26.511 | 0.163 |
| | Best (a) | 89.52 | 67.65 | 18.07 | 27.93 | | |
| | Best (b) | 89.23 | 66.75 | 23.21 | 27.88 | | |
| L1Norm-B [Li et al., 2017] | Best Benign | 90.01 | 64.89 | 19.39 | 24.52 | 26.157 | 0.146 |
| | Best (a) | 89.78 | 67.14 | 17.75 | 26.91 | | |
| | Best (b) | 89.42 | 66.66 | 21.63 | 27.55 | | |
| LRF [Joo et al., 2021] | Best Benign | 92.79 | 68.97 | 22.02 | 27.95 | 29.915 | 0.196 |
| | Best (a) | 92.46 | 70.56 | 20.91 | 28.90 | | |
| | Best (b) | 92.43 | 69.50 | 25.40 | 29.22 | | |
| NPPM [Gao et al., 2021] | Best Benign | 91.92 | 66.83 | 22.23 | 25.63 | 26.998 | 0.198 |
| | Best (a) | 91.79 | 67.56 | 22.39 | 25.91 | | |
| | Best (b) | 91.67 | 67.16 | 23.97 | 25.60 | | |
| SFP [He et al., 2018] | - | 90.28 | 66.71 | 20.35 | 25.47 | - | - |
| SR-GKP (Ours) | Best Benign | **92.21** | 66.38 | 21.91 | 25.98 | 26.717 | 0.176 |
| | Best (a) | 91.52 | **69.33** | 14.83 | 27.94 | | |
| | Best (b) | 92.04 | 66.37 | **23.55** | 25.80 | | |

Table 16: Full experiments of ResNet-56 on CIFAR-10 with pruning rate ≈ 62.5%. All pruning methods are performed on the same baseline model. "Best (a)" represents the performance of a model checkpoint that meets the showcased MACs/Params reduction and performs best against criterion (a).

| Method | Criterion | Benign | (a) $\text{FGSM}_{\varepsilon=0.01}$ | (b) $\text{FGSM}_{\varepsilon=0.1}$ | (c) $\text{PGD}_{\text{max\_iter}=3}^{\varepsilon=8/255,\ \varepsilon_{\text{step}}=2/255}$ | MACs (M) | Params (M) |
|---|---|---|---|---|---|---|---|
| Unpruned | - | 93.24 | 75.15 | 39.64 | 42.58 | 126.561 | 0.853 |
| CC [Li et al., 2021] | Best Benign | 93.57 | 73.63 | 25.30 | 35.40 | | |
| | Best (a) | 93.33 | 74.29 | 24.96 | 35.54 | 48.692 | 0.421 |
| | Best (b) | 93.37 | 62.56 | 26.12 | 35.25 | | |
| DHP [Li et al., 2020] | Best Benign | 91.66 | 70.66 | 29.75 | 31.40 | | |
| | Best (a) | 91.36 | 71.22 | 26.36 | 31.05 | - | - |
| | Best (b) | 91.48 | 70.41 | 30.15 | 31.27 | | |
| FPGM [He et al., 2019] | Best Benign | 92.64 | 71.80 | 35.17 | 35.17 | | |
| | Best (a) | 92.31 | 72.58 | 35.98 | 35.94 | - | - |
| | Best (b) | 92.62 | 71.86 | 35.99 | 35.61 | | |
| HRank [Lin et al., 2020] | - | 90.63 | 69.49 | 17.14 | 29.51 | - | - |
| L1Norm-A [Li et al., 2017] | Best Benign | 91.79 | 68.95 | 24.68 | 34.90 | | |
| | Best (a) | 91.12 | 71.76 | 37.01 | 36.97 | 47.562 | 0.355 |
| | Best (b) | 91.47 | 70.10 | 39.83 | 37.03 | | |
| L1Norm-B [Li et al., 2017] | Best Benign | 91.56 | 69.56 | 32.80 | 33.61 | | |
| | Best (a) | 90.66 | 71.24 | 33.22 | 35.09 | 47.794 | 0.322 |
| | Best (b) | 91.07 | 69.25 | 26.19 | 33.66 | | |
| NPPM [Gao et al., 2021] | Best Benign | 93.07 | 73.23 | 29.66 | 35.24 | | |
| | Best (a) | 92.84 | 74.21 | 28.27 | 35.38 | 52.550 | 0.446 |
| | Best (b) | 93.02 | 72.91 | 30.39 | 34.03 | | |
| SFP [He et al., 2018] | - | 92.24 | 72.21 | 33.65 | 35.39 | - | - |
| SR-GKP (Ours) | Best Benign | **92.93** | 70.94 | 21.15 | 32.01 | | |
| | Best (a) | 92.69 | **73.54** | 35.42 | 38.39 | 48.409 | 0.323 |
| | Best (b) | 92.69 | 73.38 | **36.53** | 38.95 | | |

Table 17: Full experiments of ResNet-110 on CIFAR-10 with pruning rate ≈ 62.5%. All pruning methods are performed on the same baseline model. "Best (a)" represents the performance of a model checkpoint that meets the showcased MACs/Params reduction and performs best against criterion (a).

| Method | Criterion | Benign | (a) $\text{FGSM}_{\varepsilon=0.01}$ | (b) $\text{FGSM}_{\varepsilon=0.1}$ | (c) $\text{PGD}_{\text{max\_iter}=3}^{\varepsilon=8/255,\ \varepsilon_{\text{step}}=2/255}$ | MACs (M) | Params (M) |
|---|---|---|---|---|---|---|---|
| Unpruned | - | 94.26 | 78.55 | 48.84 | 50.85 | 254.995 | 1.728 |
| CC [Li et al., 2021] | Best Benign | 94.29 | 73.77 | 24.50 | 36.32 | | |
| | Best (a) | 94.05 | 74.23 | 24.87 | 36.71 | 98.582 | 0.727 |
| | Best (b) | 94.03 | 73.70 | 26.03 | 36.44 | | |
| DHP [Li et al., 2020] | Best Benign | 92.73 | 71.39 | 23.19 | 35.51 | | |
| | Best (a) | 92.35 | 72.41 | 23.70 | 36.22 | - | - |
| | Best (b) | 92.50 | 71.27 | 25.13 | 34.86 | | |
| FPGM [He et al., 2019] | Best Benign | 94.11 | 76.11 | 47.62 | 47.54 | | |
| | Best (a) | 94.00 | 76.52 | 48.39 | 47.75 | - | - |
| | Best (b) | 93.93 | 76.41 | 49.33 | 47.62 | | |
| HRank [Lin et al., 2020] | - | 91.94 | 70.13 | 15.04 | 30.19 | - | - |
| L1Norm-A [Li et al., 2017] | Best Benign | 92.50 | 73.06 | 40.19 | 41.77 | 97.952 | 0.622 |
| | Best (a) & (b) | 91.51 | 74.99 | 42.62 | 43.45 | | |
| L1Norm-B [Li et al., 2017] | Best Benign | 94.04 | 74.81 | 41.82 | 41.28 | | |
| | Best (a) | 93.79 | 75.55 | 42.36 | 41.99 | 101.256 | 0.484 |
| | Best (b) | 93.86 | 74.96 | 43.43 | 41.99 | | |
| LRF [Joo et al., 2021] | Best Benign | 94.10 | 75.47 | 20.66 | 39.87 | | |
| | Best (a) | 93.88 | 76.96 | 33.44 | 42.52 | 94.479 | 0.638 |
| | Best (b) | 93.98 | 76.21 | 34.55 | 41.77 | | |
| NPPM [Gao et al., 2021] | Best Benign | 93.93 | 74.71 | 31.37 | 38.81 | | |
| | Best (a) | 93.76 | 75.32 | 31.12 | 39.26 | 99.915 | 0.746 |
| | Best (b) | 93.79 | 75.02 | 32.62 | 39.26 | | |
| SFP [He et al., 2018] | - | 92.98 | 76.08 | 52.15 | 47.26 | - | - |
| SR-GKP (Ours) | Best Benign | **94.31** | 76.31 | 43.88 | 45.44 | | |
| | Best (a) | 94.17 | **76.52** | 43.98 | 46.01 | 97.217 | 0.654 |
| | Best (b) | 94.27 | 76.17 | **44.56** | 45.74 | | |

### D.2.3  VGG-16 on CIFAR-10

Please refer to Table 18 with Figure 9 for details.

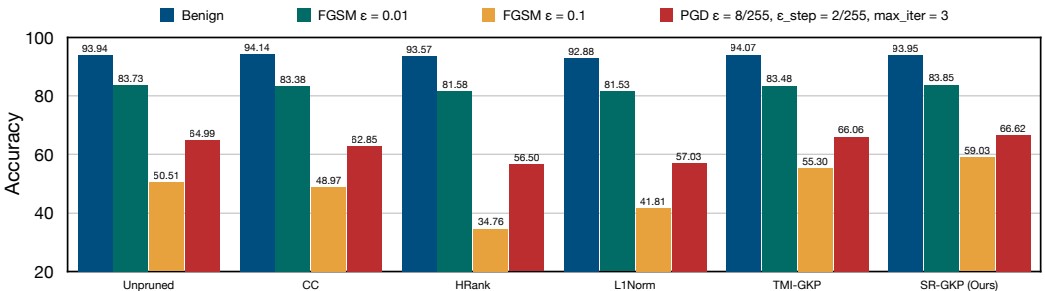

Figure 9: "Superscore" visualization of VGG-16 on CIFAR-10.

Table 18: Full experiments of VGG-16 on CIFAR-10. All pruning methods are performed on the same baseline model. "Best (a)" represents the performance of a model checkpoint that meets the showcased MACs/Params reduction and performs best against criterion (a).

| Method | Criterion | Benign | (a) FGSM$_{\varepsilon=0.01}$ | (b) FGSM$_{\varepsilon=0.1}$ | (c) PGD$_{max\_iter=3}^{\varepsilon=8/255,\ \varepsilon_{step}=2/255}$ | MACs (M) | Params (M) |
|---|---|---|---|---|---|---|---|
| Unpruned | - | 93.94 | 83.73 | 50.51 | 64.99 | 313.433 | 14.728 |
| CC [Li et al., 2021] | Best Benign | 94.14 | 83.09 | 47.83 | 62.81 | 178.107 | - |
| | Best (a) | 93.88 | 83.38 | 47.70 | 62.74 | | |
| | Best (b) | 93.97 | 83.14 | 48.97 | 62.85 | | |
| GAL [Lin et al., 2019] | - | 91.29 | 81.69 | 61.96 | 67.29 | 203.224 | 7.732 |
| HRank [Lin et al., 2020] | Best Benign | 93.57 | 81.45 | 31.80 | 56.28 | 212.264 | 8.700 |
| | Best (a) | 93.53 | 81.58 | 34.00 | 56.50 | | |
| | Best (b) | 93.54 | 81.52 | 34.76 | 56.34 | | |
| L1Norm [Li et al., 2017] | Best Benign | 92.88 | 80.85 | 37.21 | 56.43 | 179.561 | 9.135 |
| | Best (a) | 92.41 | 81.53 | 31.10 | 56.17 | | |
| | Best (b) | 92.06 | 80.48 | 41.81 | 57.03 | | |
| TMI-GKP [Zhong et al., 2022] | Best Benign | **94.07** | 83.48 | 55.30 | 66.06 | 178.184 | 8.293 |
| SR-GKP (Ours) | Best Benign | 93.95 | **83.49** | **56.64** | 65.57 | 178.184 | 8.293 |
| | Best (a) | 93.77 | **83.85** | **57.48** | **66.51** | | |
| | Best (b) | 93.86 | **83.61** | **59.03** | **66.62** | | |

#### D.2.4   ResNet-56/101 on Tiny-ImageNet

Please refer to Table 19 for details.

Table 19: Full experiments of ResNet-56/101 on Tiny-Imagenet. All pruning methods are performed on the same baseline model. "Best (a)" represents the performance of a model checkpoint that meets the showcased MACs/Params reduction and performs best against criterion (a).

| Model | Method | Criterion | Benign | (a) FGSM$_{\varepsilon=0.001}$ | (b) FGSM$_{\varepsilon=0.01}$ | (c)FGSM$_{\varepsilon=0.1}$ | (d) PGD$_{max\_iter=3}^{\varepsilon=4/255,\ \varepsilon_{step}=1/255}$ | MACs (M) | Params (M) |
|---|---|---|---|---|---|---|---|---|---|
| ResNet-56 | Unpruned | - | 55.59 | 53.55 | 28.29 | 8.00 | 15.80 | 506.254 | 0.865 |
| | SR-GKP (Ours) | Best Benign | **54.83** | **54.14** | **29.22** | **7.71** | **17.08** | 318.690 | 0.547 |
| | TMI-GKP [Zhong et al., 2022] | Best Benign | 51.48 | 50.07 | 27.23 | 7.62 | 15.42 | 318.690 | 0.547 |
| ResNet-101 | Unpruned | - | 65.51 | 65.12 | 48.10 | 10.13 | 37.66 | 10081.092 | 42.902 |
| | SR-GKP (Ours) | Best Benign | **67.21** | 66.52 | 46.98 | 8.95 | **37.34** | 5721.113 | 24.226 |
| | | Best (a) | 65.69 | **64.91** | 37.95 | 10.95 | 25.94 | | |
| | | Best (b) | 65.61 | 64.71 | **38.27** | 10.85 | 25.97 | | |
| | | Best (c) | 65.69 | 64.91 | 37.95 | **10.96** | 25.95 | | |
| ResNet-101 | TMI-GKP [Zhong et al., 2022] | Best Benign | 64.69 | 64.03 | 42.57 | 8.40 | 32.52 | 5721.113 | 24.226 |
| | | Best (a) | 63.53 | 62.23 | 33.47 | 10.46 | 20.61 | 5721.113 | 24.226 |
| | | Best (b) | 63.60 | 61.94 | 33.71 | 10.71 | 20.47 | 5721.113 | 24.226 |
| | | Best (c) | 63.50 | 61.93 | 33.67 | 10.77 | 20.63 | 5721.113 | 24.226 |

#### D.2.5   Comparison with Channel Pruned HARP

HARP by Zhao and Wressnegger [2023] is one of a few adversarially robust structured pruning methods available for comparison, as it has a channel pruning version offered in its appendix (HARP-CP). However, HARP utilizes several different training schedules than other methods, and we therefore cannot compare it with other methods using a unified baseline (as in Table 18). To fulfill the comparison, we opt to align SR-GKP with HARP under different potential training schedule settings offered in HARP literature or our experiment setup. Please refer to Table 20 for details.

Table 20: Comparison with Channel-Pruned HARP (HARP-CP) by Zhao and Wressnegger [2023] on VGG-16 with CIFAR-10 dataset. (AT = Adversarial Training; NT = Natural Training; AS = epoch with best adversarial accuracy is saved; NS = epoch with best benign accuracy is saved; LS = last epoch is saved; SS = "Superscore" readings according to Section 4.2 and Table 10.

| Method | Baseline Budget | Pruning Budget | Fine-tuning Budget | Best Benign | $FGSM_{\varepsilon=0.01}$ | $FGSM_{\varepsilon=0.1}$ | PGD |
|---|---|---|---|---|---|---|---|
| HARP-CP | NT, 300 (AS) | AT, 20 (LS) | NT, 300 (NS) | 91.66 | 55.11 | 23.13 | 3.30 |
| HARP-CP | NT, 300 (NS) | AT, 20 (LS) | NT, 300 (NS) | 87.04 | 23.49 | 10.71 | 0.05 |
| HARP-CP | NT, 300 (AS) | AT, 20 (LS) | AT, 100 (AS) | 72.06 | 64.08 | 17.96 | 50.93 |
| SR-GKP (Ours) | NT, 300, (LS) | 0 | NT, 300 (SS) | **93.95** | **83.85** | **59.03** | **66.62** |
| SR-GKP (Ours) | NT, 300, (LS) | 0 | NT, 100 (SS) | **93.48** | **82.62** | **52.35** | **62.18** |

We believe it is safe to conclude that SR-GKP showcased clear dominance over the HARP-CP.

Although one network-dataset combination is not a comprehensive evaluation, Figure 13 of HARP suggests HARP-CP is extremely close to RAP-ADMM [Ye et al., 2019] — a method we already compared — in terms of both benign and adversarial acc; and from Figure 6, 7, 8 as well as Table 3, we may conclude that RAP-ADMM performs vastly below SR-GKP on various metrics. We find it interesting that HARP and RAP-ADMM — two methods utilizing adversarial training — still perform worse than SR-GKP on most adversarial tasks, though SR-GKP only sees benign inputs; indicating the effectiveness of SR-GKP. We suspect this is mainly due to the fact HARP and RAP-ADMM are not structured pruning-focused methods.

### D.3 Ablation Studies

Please refer to Table 21, Table 22, and Table 23 for ablation studies on hyperparameter $\alpha$ in Equation 2. We denote it as CSB as it is in essence a "cost-smoothness balancer." It can be observed that a relatively high CSB — meaning giving more bias to the distance-based cost metrics — may yield better performance.

Table 21: Ablation study of cost-smoothness balancer "CSB" ($\alpha$ in Equation 2) on ResNet-32 with CIFAR-10.

| Method | Criterion | Benign | (a) $FGSM_{\varepsilon=0.01}$ | (b) $FGSM_{\varepsilon=0.1}$ | MACs (M) | Params (M) |
|---|---|---|---|---|---|---|
| Unpruned | - | 92.80 | 71.93 | 31.35 | 69.479 | 0.464 |
| 0.1 CSB | Best Benign | 92.79 | 70.39 | 28.09 | 39.545 | 0.263 |
| | Best (a) | 92.59 | 71.50 | 29.03 | | |
| | Best (b) | 92.47 | 70.94 | 29.69 | | |
| 0.25 CSB | Best Benign | 93.01 | 70.45 | 27.91 | 39.545 | 0.263 |
| | Best (a) | 92.70 | 71.32 | 30.46 | | |
| | Best (b) | | | | | |
| 0.5 CSB | Best Benign | 93.07 | 68.74 | 28.07 | 39.545 | 0.263 |
| | Best (a) | 92.54 | 70.63 | 21.42 | | |
| | Best (b) | 92.78 | 70.29 | 30.47 | | |
| 0.75 CSB | Best Benign | 92.93 | 70.55 | 30.02 | 39.545 | 0.263 |
| | Best (a) | 92.70 | 71.09 | 30.89 | | |
| | Best (b) | | | | | |
| **0.9 CSB** | Best Benign | 92.97 | 70.57 | 29.31 | 39.545 | 0.263 |
| | Best (a) | 92.88 | 71.52 | 30.39 | | |
| | Best (b) | 92.86 | 70.79 | 31.32 | | |

Table 22: Ablation study of cost-smoothness balancer "CSB" ($\alpha$ in Equation 2) on ResNet-56 with CIFAR-10.

| Method | Criterion | Benign | (a) FGSM$_{\varepsilon=0.01}$ | (b) FGSM$_{\varepsilon=0.1}$ | MACs (M) | Params (M) |
|---|---|---|---|---|---|---|
| Unpruned | - | 93.24 | 75.15 | 39.64 | 126.561 | 0.853 |
| 0.1 CSB | Best Benign | 93.72 | 75.17 | 39.90 | 71.855 | 0.482 |
|  | Best (a) | 93.38 | 75.79 | 42.53 |  |  |
|  | Best (b) |  |  |  |  |  |
| 0.25 CSB | Best Benign | 93.83 | 75.68 | 40.40 | 71.855 | 0.482 |
|  | Best (a) | 93.70 | 76.49 | 42.35 |  |  |
|  | Best (b) | 93.57 | 75.72 | 43.92 |  |  |
| 0.5 CSB | Best Benign | 93.76 | 75.61 | 41.20 | 71.855 | 0.482 |
|  | Best (a) | 93.41 | 76.69 | 42.58 |  |  |
|  | Best (b) | 93.45 | 75.97 | 42.69 |  |  |
| **0.75/0.9 CSB** | Best Benign | 94.08 | 75.89 | 42.60 | 71.855 | 0.482 |
|  | Best (a) | 93.83 | 76.40 | 45.17 |  |  |
|  | Best (b) |  |  |  |  |  |

Table 23: Ablation study of cost-smoothness balancer "CSB" ($\alpha$ in Equation 2) on ResNet-110 with CIFAR-10.

| Method | Criterion | Benign | (a) FGSM$_{\varepsilon=0.01}$ | (b) FGSM$_{\varepsilon=0.1}$ | MACs (M) | Params (M) |
|---|---|---|---|---|---|---|
| Unpruned | - | 94.26 | 78.56 | 48.83 | 254.995 | 1.728 |
| 0.1 CSB | Best Benign | 94.60 | 78.44 | 44.20 | 144.551 | 0.976 |
|  | Best (a) | 94.47 | 79.62 | 46.03 |  |  |
|  | Best (b) | 94.35 | 79.41 | 47.5 |  |  |
| 0.25 CSB | Best Benign | 94.50 | 77.86 | 47.07 | 144.551 | 0.976 |
|  | Best (a) | 94.35 | 78.65 | 46.04 |  |  |
|  | Best (b) | 94.40 | 78.52 | 48.10 |  |  |
| **0.5 CSB** | Best Benign | 95.00 | 78.01 | 46.53 | 144.551 | 0.976 |
|  | Best (a) | 94.69 | 79.14 | 47.49 |  |  |
|  | Best (b) | 94.60 | 78.92 | 49.09 |  |  |
| 0.75 CSB | Best Benign | 94.49 | 77.97 | 45.93 | 144.551 | 0.976 |
|  | Best (a) | 94.26 | 78.91 | 47.84 |  |  |
|  | Best (b) | 94.32 | 78.44 | 48.89 |  |  |
| 0.9 CSB | Best Benign | 94.54 | 78.21 | 47.16 | 144.551 | 0.976 |
|  | Best (a) | 94.29 | 78.81 | 46.71 |  |  |
|  | Best (b) | 94.31 | 78.25 | 48.55 |  |  |

