# OpenReview forum: "One Less Reason for Filter Pruning: Gaining Free Adversarial Robustness with Structured Grouped Kernel Pruning"
_NeurIPS.cc/2023/Conference — NeurIPS 2023 poster_

### Official Review · Reviewer_rEnW · 2023-06-26

**Soundness:** 2 fair
**Presentation:** 2 fair
**Contribution:** 3 good
**Rating:** 6
**Confidence:** 3

**Summary:**

The authors base their work on top of Grouped Kernel Pruning (GKP) to alleviate the issue of adversarial robustness when pruning CNNs structurally. The authors propose Smoothly Robust Grouped Kernel Pruning (SR-GKP) that alters how the filter grouping occurs in GKP and how the grouped kernels are selected. The authors claim that this method is one-shot and data-free. Post one-shot pruning, the authors fine-tune the pruned models on the original training set. The authors claim that this method has no extra cost as opposed to other pruning methods to gain adversarial robustness.

**Strengths:**

S1. The problem considered is important

S2. The method is clever and technically sound.

S3. Empirical evidence shows merit in using this approach.

**Weaknesses:**

W1. The baselines for the ImageNet experiments are weak (Table 3). Some suggestions for incorporating recent baselines: [a], [b], [c] (GitHub repos are available).

W2. Lack of ablation studies w.r.t. $\alpha$

W3. Lack of adherence to the NeurIPS 2023 template (Figure 2)

W4. Writing needs to be improved.
Some typos:
- Line 193. find
- Line 222 and 243. (Appendix C) [brackets]
- Line 248. Did you mean tamper?
- Line 310. (and similar experiments showcased in ??)








[a] Only Train Once: A One-Shot Neural Network Training And Pruning Framework. Chen et al. 2021

[b] Neural Pruning via Growing Regularization. Wang et al. 2021

[c] Structural Pruning via Latency-Saliency Knapsack. Shen et al. 2022

[d] DFPC: Data flow driven pruning of coupled channels without data. Narshana et al. 2023

**Questions:**

Q1. As stated on lines 276-277 you state that you fine-tune the model. However, you state that your method is data-free. Why?

Q2. When discarding grouped kernels (Stage 2), how do you decide when to stop pruning a layer?

Q3. How do you decide an appropriate value of $\alpha$? Does the optimal $\alpha$ value remain the same across datasets and architectures?

Q4. When pruning a layer (structured), you essentially reduce the output tensor size. This affects the kernels in the following layers (they also need to be pruned). Do you include them to decide whether a grouped kernel is to be pruned?

**Limitations:**

The method makes progress in alleviating the issue of adversarial robustness. However, the pruned models are still prone to suffer from adversarial inputs, as shown in the empirical results.

---

> ### Author Rebuttal · Authors · 2023-08-10
>
> (No char left, but thx!)
>
> ### **[W1 - Weak baseline on ImageNet]: The choice of compared methods on ImageNet is derived from our CIFAR10 experiments. We didn't cherry/lemon pick them.**
>
> Given we need to replicate full ImageNet experiments to obtain adv. results, we opt only to compare methods showcased strong performance in CIFAR10 exps, which are SFP and FPGM ([Table {9, 10, 11}](https://openreview.net/attachment?id=Pjky9XG8zP&name=supplementary_material#page=18)). It just so happened that both methods have weak benign readings on ImageNet, which suggests it is hard to deliver strong results across different dataset-model combinations, making SR-GKP's strong across-the-board performance particularly cherished.
>
> We believe this method-filtering mechanism is sound. Yet, we want to note further that ImageNet with adv. metrics are rarely evaluated under the context of pruning + adv. robustness [1, 2], mainly because many of ImageNet subclasses are closely related in terms of semantics (e.g., man-eating shark vs tiger shark) [3]. We provide the ImageNet results primarily to show SR-GKP can deliver competitive benign ImageNet performance. We will emphasize this in our main text to add clarity.
>
> That being said since the reviewer is interested, we incorporated those methods on ImageNet and CifarResNets to the best of their repo's reproducibility:
>
> | Method on ResNet50-ImageNet | Benign | FGSM $\varepsilon = 0.001$ | FGSM $\varepsilon = 0.01$ | FGSM $\varepsilon = 0.1$ | PGD |
> |:-:|:-:|:-:|:-:|:-:|:-:|
> | Unpruned | 76.13 | 70.18 | 28.70 | 13.93 | 9.27 |
> | OTOv2 (post-train) | **75.38** | **68.04** | 21.01 | 12.76 | 5.26  |
> | DFPC | 73.80 | 67.45 | 25.15 | 12.24 | 7.10 |
> | FPGM | 75.04 | 68.50 | 25.84 | 13.06 | 7.43 |
> | SFP | 58.50 | 55.72 | 25.82 | 9.01 | 11.00 |
> | SR-GKP | 75.34 | **68.04** | **26.65** | **15.94** | 7.85 |
>
> | Method | Model on CIFAR10 | Benign | FGSM $\varepsilon = 0.01$ | FGSM $\varepsilon = 0.1$ | PGD |
> |:-:|:-:|:-:|:-:|:-:|:-:|
> | OTOv2 (from-scratch) | ResNet32  | 90.97 | 66.36 | 17.28 | 27.15 |
> | OTOv2 (post-train)   | ResNet32  | 92.14 | 70.63 | 28.01 | **31.96** |
> | SR-GKP               | ResNet32  | **93.02** | **71.48** | **31.11** | 31.63 |
> | | | | | | |
> | OTOv2 (from-scratch) | ResNet56  | 91.57 | 65.91 | 23.63 | 25.52 |
> | OTOv2 (post-train)   | ResNet56  | 93.02 | 74.29 | 40.46 | 41.58 |
> | GReg-1               | ResNet56  | 91.06 | 70.56 | 23.66 | 33.48 |
> | SR-GKP               | ResNet56  | **94.08** | **76.40** | **45.17** | **44.85** |
> | | | | | | |
> | OTOv2 (from-scratch) | ResNet110 | 91.58 | 71.43 | 23.94 | 34.74 |
> | OTOv2 (post-train)   | ResNet110 | 93.99 | 77.25 | 47.96 | 48.46 |
> | SR-GKP               | ResNet110 | **95.00** | **79.14** | **49.09** | **50.01** |
>
> Some asked methods are not listed due to reproduction hardship: e.g., GReg induces memory overflow on our 80G A100 GPUs; DFPC's data-driven implementation is not yet shared; HALP requires a handcrafted config and hardware-specific latency LUT. Due to character limitation we will leave it as that; we'd be happy to provide more info during the discussion session, should the reviewer be interested.
>
> ### **[W2, Q3 - Lack of ablation study on $\alpha$]: It is in Appendix D.3**
>
> We kindly direct the reviewer's attention to [Appendix D.3](https://openreview.net/attachment?id=Pjky9XG8zP&name=supplementary_material#page=22), where there is a full-scale ablation study on $\alpha$. We'd say the choice of $\alpha$ is pretty consistent.
>
> ### **[W3&4 Template and typos]: Thanks, will fix!**
>
> ---
>
> ### **[Q1 - Why claim "data-free" when fine-tuning is required?]: Ambiguous term — good point, will replace.**
>
> We meant the pruning stage of SR-GKP can be done without access to data. Upon inspection, it looks like some pruning arts use the term "data-free" (no access to data) and "data-agnostic" (independent of data) interchangeably. e.g., SynFlow [4], a data-agonistic method, is referred as data-free in [5]. [6], a namely "data-free" method, still *"enabling operations like fine-tuning..."*.
>
> To avoid ambiguities, we will replace it with "weight-dependent" should there be a camera-ready version, as we believe the reviewer's perspective is less ambiguous and appears to be more mainstream in recent pruning arts [DFPC, 7].
>
> ### **[Q2 - When to stop pruning a layer?]: When it reaches the target (universal) pruning ratio.**
>
> ### **[Q4 - SR-GKP reduces output tensor size between layers]: Not really.**
>
> SR-GKP does not reduce the output tensor size. Rather, it reduces the computation required to generate those (between-layer) outputs. Take [Figure 1](https://openreview.net/attachment?id=Pjky9XG8zP&name=supplementary_material#page=2) as an example: the unpruned model has 6 features maps as output for having 6 filters. Yet in GKP, it still has 6 feature maps as it has 3 grouped filters each with a size of 2; but fewer kernels (3, vs 6) are utilized to generate such feature maps.
>
> ### **[Limitation - SR-GKP pruned models still suffer from adv. inputs]: True, but SR-GKP already showcased better performance than unpruned baseline, as well as many methods rely on adv. training.**
>
> Given an unlimited perturbation budget, no adv. defense can be immune to all adv. attacks, regardless if the victim model is pruned or not. We consider our method a worthy progression for the subtitled reasons, yet the investigation-side of our work also unveils many previously unknown findings to the pruning community.
>
> ---
>
> [1] Chen and Zhang et al., Linearity Grafting... Robustness. ICML 2022
> [2] Ye et al., Adversarial Robustness ... Both? ICCV 2019
> [3] Ozbulak & Pintor et al., Evaluating Adversarial ... Classes. NeurIPS 2021 (Workshop)
> [4] Tanaka & Kunin et al., Pruning neural networks... Flow. NeurIPS 2020
> [5] Pellegrini et al., Neural Network Pruning Denoises ... Tasks. PMLR 2022
> [6] Srinivas et al., Data-free Parameter ... Networks. arXiv 2015
> [7] Yvinec et al., RED: Looking for Redundancies ... Networks. NeurIPS 2021

---

> > ### Author Response · Authors · 2023-08-18
> > **An invite to discussion; as well as additional/updated results on GReg.**
> >
> > Right now, with `7554`, you are the only reviewer staying on the negative side of the rating, and we would like to ensure we have adequately addressed your concerns. We believe it is fair to say — despite a rating of `4` — that **your feedback is plenty positive: as you recognize our work to score the trifecta of *"problem being important, method being technically sound, and evaluation to be supportive."*** We appreciate your recognition.
> >
> > Your main concerns are two-fold: some experiment requests on ImageNet & hyperparameters $\alpha$, plus some cosmetic suggestions regarding typos and formatting. We thank you for your careful read and will undoubtedly address the cosmetic issues. Yet, we believe we have clearly justified why we did our ImageNet experiments the way we showed and delivered your asked methods to the best of their repo's reproducibility. Regarding $\alpha$, as mentioned above, we have already conducted such experiments in our appendix ([D.3](https://openreview.net/attachment?id=Pjky9XG8zP&name=supplementary_material#page=22)) and will highlight them in our updated main text.
> >
> > **We argue we have properly addressed your concerns (and also questions), as they are all very factually rooted, and we would like to invite you to have a discussion should the opportunity permits.**
> >
> > ---
> >
> > In our rebuttal, we reported your asked methods' performance on some BasicBlock CifarResNets; for one, to showcase more results and, two, to demonstrate why they won't cut to the ImageNet evaluation against our screening mechanism. **We noticed the GReg-1 on ResNet-56 readings seem to be inconsistent with their paper's report** ([93.06% reported](https://openreview.net/pdf?id=o966_Is_nPA#page=6) vs 91.06% replicated benign acc, even though they are on a close-enough pruning ratio with more epoch budget granted to the replication run for a fair comparison). Upon inspection, it looks like we haven't got the config quite right; with the proper setting, we'd have the following results (reported with additional details).
> >
> > | ResNet56 on CIFAR10 | Benign | FGSM $\varepsilon = 0.01$ | FGSM $\varepsilon = 0.1$ | PGD | MACs(M) | Params(M) |
> > |:-|:-:|:-:|:-:|:-:|:-:|:-:|
> > | Unpruned Baseline for GReg | 93.90 | 75.47 | 38.80 | 42.81 | 126.551 | 0.853 |
> > | Unpruned Baseline for the rest |  93.24 | 75.15 | 39.64 | 42.58 | 126.561 | 0.853
> > ||
> > | OTOv2 (from-scratch)  | 91.57 | 65.91 | 23.63 | 25.52 | 79.780 | 0.480 |
> > | OTOv2 (post-train)    | 93.02 | 74.29 | 40.46 | 41.58 | 66.196 | 0.554 |
> > | *[UPDATED]* GReg-1    | 93.30 | 74.78 | 37.11 | 42.22 | 76.143 | 0.552 |
> > | *[NEW]* GReg-2        | 93.74 | 75.40 | 39.30 | 43.71 | 76.143 | 0.552 |
> > | SR-GKP                | **94.08** | **76.40** | **45.17** | **44.85** | 71.855 | 0.482 |
> >
> > Note this time we added GReg-2  — a more faithful representation of the GReg algorithm in its full force, which is also empirically stronger than GReg-1 — even though it is not reported with ResNet-56 in its original paper. **With the added and updated results, we may still observe a significant gap between GReg/OTO-variants and SR-GKP, indicating our method's effectiveness.**

---

> > > ### Comment · Reviewer_rEnW · 2023-08-18
> > > **Thank you for your engaging responses.**
> > >
> > > Your rebuttal addresses my concerns. Your responses were impressive. However, I have two follow ups.
> > > 1. [Reg. your response to Q4.] So you basically use "structured" masks to prune the network? And there is no wall-clock inference latency speedup in its current implementation?
> > > 2. [Reg. your response W1] I believe that providing additional baselines for ImageNet would only strengthen your work (which indeed is the case) rather than trying to maintain the same set of baselines for CIFAR and ImageNet experiments.
> > >
> > > I am willing to increase the score once the writing issues pointed out by reviewers Tjyo, sejQ (one-shot disambiguation), xyz8, and myself (typos + usage of "data-free") are incorporated into the manuscript.

---

> > > > ### Author Response · Authors · 2023-08-18
> > > > **Addressing follow-up questions & comments. Thank you for willing to increase the score!**
> > > >
> > > > We are glad that you find our response helpful, here we address your additional questions and comments.
> > > >
> > > > ### **[Follow-up #1 - Is GKP implemented as "structured" masks?]: No, it is densely structured.**
> > > >
> > > > This is a good question. **The short answer is no, SR-GKP is not implemented with masks, and we can observe direct inference speed up in the attached table below.** But a more faithful answer is a bit more convoluted, as it involves some engineering choices between different scholars and methods, as well as some framework optimization developments.
> > > >
> > > >
> > > > So in the realm of structured pruning, there are two types of popular implementations in practice:
> > > >
> > > > 1. **To deploy a structured binary mask upon the original unpruned model**, but clear the gradients of some (or all) pruned components before the weight update step during fine-tuning, therefore theoretically "structural pruned" the model. In this implementation, the pruned model will not reduce in dimension; **thus, no acceleration and compression benefits can be directly observed.**
> > > >     * This is a popular implementation — as surveyed in [Table 8 - "Zero-Masked?" column](https://openreview.net/attachment?id=Pjky9XG8zP&name=supplementary_material#page=18) — because some iterative pruning methods would like the pruned/zeroed components to reactive during fine-tuning, then prune another set of components instead. If the pruning granularity is structured (e.g., filter-level), the fine-tuned model can be converted to #2 by the end of update.
> > > > 2. **To remove the pruned components entirely**, where the pruned model will reduce in size, **providing immediate compression benefits.**
> > > >
> > > > **Our method, SR-GKP, is implemented as #2. Meaning that if you inspect the weights of a GKP-pruned network, they are in regular shapes, and there is no zeroed weight to be found** — a.k.a. "densely structured" ([Line 57-61](https://openreview.net/attachment?id=Pjky9XG8zP&name=supplementary_material#page=2)). We find your question pretty insightful, as the main contribution of early GKP works like [1] is they leveraged the grouped convolution format to remove the need for zero-padding (which is traditionally required for kernel-level pruning) and therefore enabled a finer densely structured granularity.
> > > >
> > > > In terms of inference acceleration, it is indeed the case that the previous grouped convolution implementation cannot provide speed-up benefit against standard convolution (even with much fewer MACs/Params), because the standard convolution operator has been extensively optimized, while the grouped one hasn't [2]. But after `torch 2.0`, this is not the case *at large*. For simplicity, here we compare the forward wall-clock between a standard `Conv2d` of `(C_in = 512, C_out = 512, kernel_size = (3, 3))` with the exact same `Conv2d` but with `groups = 2` (meaning half of its kernels are structurally removed).
> > > >
> > > > > Inference wall-clock comparison between a standard conv op and GKP conv op (w/ `groups=2` and `pruning_rate = 0.5`). Input size set as `(64, 3, 224, 224)`.
> > > >
> > > > | Operator | Forward  | MACs | Params |
> > > > |:--|:-:|:-:|:-:|
> > > > | Unpruned standard Conv | 129.56 ms | 550528 | 359296 |
> > > > | GKP-pruned Conv | 72.96 ms (56.31%) | 275264 (50%) | 179648 (50%) |
> > > >
> > > > **We'd say the inference speed-up is significant.** We also note that standard GKP can be easily combined with filter pruning (so it will both reduce between-layer output tensor size, as well as the compute cost to generate those outputs). We opt not to do this as we want to align our method with TMI-GKP as much as possible, so the performance gaps can be attributed to our *Smoothness Snaking* and *Smooth Beam Greedy* design, but not some architecture differences.
> > > >
> > > > (Note, we emphasized *at large* above because we indeed can find shapes that are slower with grouped conv in `torch 2.0`. We believe this is very much a framework optimization issue that is beyond the scope of our paper and shall be on the roadmap of the torch community (as well as other more inference-specific frameworks), granted their already observed improvements.)
> > > >
> > > > **If you find the above discussion beneficial, please let us know, and we will incorporate a similar discussion in our updated appendix.**
> > > >
> > > > ### **[Follow-up #2 - More ImageNet would only strengthen our work (which indeed is the case)]: We agree, thank you for suggesting.**
> > > >
> > > >
> > > > ### **[Additional comment on cosmetic issues]: We will fix them once we can update the PDF.**
> > > >
> > > > We authors appreciate your careful read of our paper, as well as other reviewers' comments. However, per [NeurIPS 23 Reviewing/Discussion process rules](https://neurips.cc/Conferences/2023/PaperInformation/NeurIPS-FAQ), we cannot update the PDF (nor is it allowed to attach it with an external link) until camera-ready. **We hope the reviewer may consider updating the rating now, given the conference mechanism is not something we the authors can alter.**
> > > >
> > > > ---
> > > >
> > > > [1] Zhong et al., Revisit Kernel ... Convolutions. ICLR 2022
> > > > [2] torch issue #10229 and #18631.

---

> > > > > ### Comment · Reviewer_rEnW · 2023-08-18
> > > > > **Updated the score.**
> > > > >
> > > > > Thank you for your response. Your responses to follow up questions were very insightful. I have updated the score.
> > > > >
> > > > > The discussion on implementation was indeed useful. I believe it would benefit the community at large if included in the appendix of the camera-ready version.

---

> > > > > > ### Author Response · Authors · 2023-08-18
> > > > > > **Thanks again. We will incorporate such discussion.**
> > > > > >
> > > > > > Thank you for your prompt reply and score update! We will incorporate a similar discussion under the current [Appendix D.1.3 - Details of Compared Methods](https://openreview.net/attachment?id=Pjky9XG8zP&name=supplementary_material#page=17), alongside with the above mentioned [Table 8](https://openreview.net/attachment?id=Pjky9XG8zP&name=supplementary_material#page=18), where the implementations of compared methods are surveyed.

---

### Official Review · Reviewer_xyz8 · 2023-07-04

**Soundness:** 2 fair
**Presentation:** 2 fair
**Contribution:** 1 poor
**Rating:** 7
**Confidence:** 4

**Summary:**

This paper investigates the adversarial robustness of modern structured pruning methods in deep neural networks. The paper insists that while naively pruned networks show poor performance under simple adversarial attacks, there is a lack of thorough investigation into the adversarial performance of carefully designed modern structured pruning methods. The authors aim to address this gap and provide remedies named Smoothly Robust Grouped Kernel Pruning (SR-GKP), a one-shot post-training pruning method for improving adversarial robustness without incurring significant additional costs.

**Strengths:**

1. Clarity: Clear writing. Easy to understand paper.
2. The paper is technically sound.

**Weaknesses:**

1. The novelty of the method is not high. The method is highly dependent on TMI-GKP [Ref_1]. The method looks very similar to previous TMI-GKP. Even if the paper has a method with smoothness, it still sounds like one of the variants of TMI-GKP under the same umbrella.
2. The adversarial performance has an incremental improvement compared to original TMI-GKP in Table 2. Both best benign performance and best(a) performance have a small improvement (less than 2%), which is not a significant contribution.
3. The paper is based on empirical study. To increase the value of the paper, it would be better if the authors can provide a theoretical analysis on why the proposed method is helpful to both benign accuracy and adversarial accuracy (especially, why benign accuracy is increased?).
4. Typos and mistakes make readers confused.
Table 2 caption: ResNet-56 on CIFAR-10. All pruning are performed on the same baseline model. “Best (a)” represents the performance of a model checkpoint that meats
Line 263: So Smoothness(GKbranch) would suggest this particular GKbranch has a greater smoothness value (a.k.a. “less smooth”) then all other GKbranch candidates in considerations.
Line 314: Figure ?? provides a vivid illustration with LRF — a 2021 method— showing the weakest adversarially robustness across the plotted methods

---

**Post-Rebuttals**

I would like to thank authors for their detailed response and for their effort in improving paper according to reviewer’s suggestions. After reading the wall-clock runtime comparison in Table 7 pointed out by authors in supplementary material, I respectfully agree with the author’s opinion on the novelty of the paper. The authors have adequately addressed most of my concerns regarding clarity except for the theoretical analysis. I have updated my rating to accept for this paper through author-reviewer discussion.

**Questions:**

1. What is the best(a) and best(b) in Table 2? No explanations for this.
2. Why does the RAP-ADMM performance against FGSM in Figure 5 lower than the unpruned version?

**Limitations:**

Yes, the authors have addressed the limitations and potential negative societal impact of their work.

---

> ### Author Rebuttal · Authors · 2023-08-10
>
> We thank the reviewer for your feedback.
>
> ### **[W1 - Limited novelty]: Please refer to our [general comment](https://openreview.net/forum?id=Pjky9XG8zP&noteId=1EzOrtnZm3).**
>
> ### **[W2 - Incremental improvement to TMI-GKP]: We respectfully disagree.**
>
> We respectfully disagree with the reviewer's assessment. Below is an abbreviated table with SR-GKP's performance noted in comparison to TMI-GKP, we'd argue the gaps are substantial on many metrics.
>
> > Gap between SR-GKP and TMI-GKP on CIFAR10 experiments in %, `+` indicates SR-GKP is better.
>
> | Model-Dataset | Best Benign | Best FGSM $\varepsilon = 0.01$ | Best FGSM $\varepsilon = 0.1$ | Best PGD |
> |:-:|:-:|:-:|:-:|:-:|
> | ResNet32-CIFAR10        | -0.02 | **+0.91** | **+0.81** | **+1.31** |
> | ResNet56-CIFAR10        | +0.13 | +0.52 | **+1.08** | +0.34 |
> | ResNet110-CIFAR10       | +0.10 | **+1.12** | **+1.94** | **+1.01** |
> | VGG16-CIFAR10           | -0.12 | +0.37 | **+3.73** | +0.56 |
>
> > Gap between SR-GKP and TMI-GKP on ImageNet/TinyImageNet experiments in %, `+` indicates SR-GKP is better.
>
> | Model-Dataset | Best Benign | Best FGSM $\varepsilon = 0.001$ | Best FGSM $\varepsilon = 0.01$ | Best FGSM $\varepsilon = 0.1$ | Best PGD |
> |:-:|:-:|:-:|:-:|:-:|:-:|
> | ResNet56-TinyImageNet   | **+3.35** | **+4.07** | **+1.99** | **+0.09** | +1.66 |
> | ResNet101-TinyImageNet  | **+2.52** | **+2.68** | **+4.56** | +0.19 | **+4.82** |
> | ResNet50-ImageNet       | +0.27 | +0.4 | +0.59 | +0.01 | **+0.82** |
>
> Further, if we may kindly direct your attention to Table 7 ([Page 16](https://openreview.net/attachment?id=Pjky9XG8zP&name=supplementary_material#page=16)), our method achieves similar benign performance to TMI-GKP, but with a much faster wall-clock runtime on the pruning stage. Given TMI-GKP and SR-GKP share the same fine-tuning stage, our method is more "executionally efficient" overall.
>
> ### **[W3 - Need theoretical analysis]: To the best of our knowledge, nothing direct is possible, but here is a discussion.**
>
> Most provable justifications for NN often require a setup distant from the actual models used in applied tasks. e.g., two-layer ReLU-activated ultra-wide MLP [1] vs ResNets. Further, given the procedure of GKP adjusts the compute graph of the original model (standard conv —> grouped conv), theoretical analysis is particularly hard.
>
> That being said, there is literature — following the style of "principled empirical evaluations" — able to provide some insights under the general context of pruning, smoothness, and adversarial robustness. Here's a brief list:
>
> 1. [2, 3] Suggest dropout — a long-standing pruning operation — may reduce over- and under-fitting by preventing over co-adaptation and making the model's gradient more aligned. [4] touches on the seemingly contradicting claim of pruning-improving-generalization and over-parameterization double decent phenomenon and showcases that pruning may induce a generalization-favored regularization effect with a proper sparsity.
> 2. [5, 6] Suggest NNs often learn LFC first, then the HFC. Given LFC are often more robust features, a model generated by post-train pruning methods might retrain more learning on LFC, thus, leaving more room to fit on HFC and cause worse adversarial performance.
> 3. [7, 8] Showcase smoothness may reduce fitting on HFC.
>
> We believe the reviewer would agree that this kind of investigation would require a sophisticated experiment chain that largely deviated from the setup of "measuring practical performances of methods under a standard input setting" (which is the main theme of our work's experiments), and worthy of a paper of its own. Thus, we leave such kinds of studies to future works.
>
> ### **[W4 - Typos]: Thanks for the careful read, will fix!**
>
> ---
>
>
>
> ### **[Q1 - Caption clarification]**
>
> “Best (a)” represents a valid model checkpoint (that meats the showcased MACs/Params reduction) with the best performance against eval metrics (a). In this case, (a) means FGSM w/ $\varepsilon=0.01$. This is a render error, which we attempted to make a remedy by supplying a correction at the top of our appendix ([Page 13](https://openreview.net/attachment?id=Pjky9XG8zP&name=supplementary_material#page=18)). We apologize for the confusion caused.
>
> ### **[Q2 - Why do RAP-ADMM FGSM readings lower than the unpruned baseline]**
>
> We don't quite grasp the foundation of this question, as it is common for a pruned model to exhibit lower acc reading than its unpruned counterpart (regardless the task is adversarial or not), simply because the pruned network has less capacity. Particularly to [Figure 5](https://openreview.net/attachment?id=Pjky9XG8zP&name=supplementary_material#page=8), methods like CC and LRF has lower FGSM reading than the unpruned baseline.
>
> We suspect the reviewer meant to ask why RAP-ADMM has a much higher PGD performance than its unpruned baseline, as that is indeed an outlier behavior that might have stirred the reviewer's interest. This is simply because RAP-ADMM does adversarial training on PGD-perturbed input ([Line 304-309](https://openreview.net/attachment?id=Pjky9XG8zP&name=supplementary_material#page=9)).
>
> ---
>
>
> [1] Du & Zai et al., Gradient Descent Provably Optimizes Over-parameterized Neural Networks. ICLR 2019
> [2] Srivastava et al., Dropout: A Simple Way to Prevent Neural Networks from Overfitting. JMLR 2014
> [3] Liu et al., Dropout Reduces Underfitting. arXiv 2023
> [4] Jin et al., Pruning’s Effect on Generalization Through the Lens of Training and Regularization. NeurIPS 2022
> [5] Xu et al., Frequency Principle: Fourier Analysis Sheds Light on Deep Neural Networks. ICLR 2020
> [6] Wang et al., What do neural networks learn in image classification? A frequency shortcut perspective. ICCV 2023
> [7] Wang et al., High Frequency Component Helps Explain the Generalization of Convolutional Neural Networks. CVPR 2020
> [8] Wang et al., Smooth Kernels Improve Adversarial Robustness and Perceptually-Aligned Gradients. ICLR 2020 Submission

---

> > ### Author Response · Authors · 2023-08-17
> > **Thank you for raising the score. We are glad to convince you in terms of the novelty of our method.**
> >
> > Novelty, or lack thereof, is often regarded as one of the hardest criticism to refute, as it is usually the case of *"one man's vulgarity is another's lyric."* Our initial feedback is very much an incarnation of such saying: with some reviewers recognized our work to be a principled progression of existing works that *"effectively combines two existing works, and provides solid gains,"* while some viewed the solution we landed on — though backed by investigatory justifications — is potentially not technically unique enough as it too rooted in exiting works; mirroring opposite views on the very same subject matter. We heard from both sides and provided our justification [here](https://openreview.net/forum?id=Pjky9XG8zP&noteId=1EzOrtnZm3).
> >
> > **We are grateful that our [Table 7](https://openreview.net/attachment?id=Pjky9XG8zP&name=supplementary_material#page=16)** (showcasing significant wall-clock efficiency of SR-GKP over TMI-GKP, while achieving similar benign & better adversarial performance) **convinced you to recognize the novelty of our work, and we'd like to borrow this chance to direct other reviewers' attention to such material**, should they also find it relevant. Given your feedback, we will take note to emphasize it in our updated main text.
> >
> > We also agree with the reviewer's assessment that **the theoretical aspect of our paper is lacking.** Though, we believe this is very much the norm of the (structured) pruning field — e.g., to the best of our knowledge, no work is able to provably justify the efficacy of filter pruning under the train-prune-finetune pipeline, even though filter pruning has been introduced for almost a decade with vast popularity attracted [1, 2]. **Adding complications like adversarial attacks and grouped convolutions, we believe providing a faithful theoretical justification for SR-GKP is, unfortunately, beyond the currently available instruments.**
> >
> > ---
> > [1] Zhou et al., Less is More: Towards Compact CNNs. ECCV 2016
> > [2] Li et al., Pruning Filters for Efficient ConvNets. ICLR 2017

---

> > > ### Comment · Reviewer_xyz8 · 2023-08-19
> > > **Additional Question**
> > >
> > > While the reviewer has read all of the reviewers' comments and rebuttals, the reviewer would like to throw one more question about training computation time in accordance with computation analysis of HARP [1] (of which Table 14 in Appendix ). In this table, it is shown that HARP seems effective and also MAD [2] are the best effective training scheme rather than others. For the full completeness of this paper's experiment, the reviewer recommends the authors to add this perspective discussion with HARP and MAD (if possible, the more baselines would be better to compare each other).
> > >
> > >
> > > ---
> > >
> > > *References*
> > >
> > > [1] Holistic Adversarially Robust Pruning, ICLR 2023
> > >
> > > [2] Masking Adversarial Damage: Finding Adversarial Saliency for Robust and Sparse Network, CVPR 2022

---

> > > > ### Author Response · Authors · 2023-08-19
> > > > **Addressing training/fine-tuning budget and comparison to HARP.**
> > > >
> > > > ### **[Follow-up #1: Clarification on training scheme]: Sure!**
> > > >
> > > > We thank the reviewer for your careful read. First, we like to kindly direct your attention to [Section 4.1 - Experiment Setups](https://openreview.net/pdf?id=Pjky9XG8zPl#page=7) and [Appendix D.1.1 - Details of Experiment Setups](https://openreview.net/attachment?id=Pjky9XG8zP&name=supplementary_material#page=17), where we indicate our fine-tuning/retraining budget. For your convenience, here we summarize such info: on all CIFAR experiments, the SR-GKP pruned network is fine-tuned for 300 epochs on standard SGD, with a `StepLR(step_size=30, gamma=0.1)`, and an initial lr of 0.01. The baseline training scheme is identical to the fine-tuning one, except having an initial lr of 0.1. We opt for this setup as it is utilized in TMI-GKP [1], as well as in many other pruning literature [2, 3].
> > > >
> > > > So if we'd format it in the style of HARP Table 14, it'd look like this:
> > > >
> > > > | Method | Multi-Stage Pipeline | Pretraining | Pruning | Fine-tuning | Total |
> > > > |---|:-:|:-:|:-:|:-:|:-:|
> > > > | SR-GKP | Yes | 300  | 0  | 300 | 600 epochs |
> > > >
> > > >
> > > > We recognize this is a much higher budget than HARP's (220 total epochs). However, we'd note that 1) HARP does PGD-perturbed adv. training, which is much more expensive/slower than SR-GKP's training on benign inputs; 2) Reviewer `sejQ` has also requested a comparison with HARP — so good suggestion on your side! —  which we delivered [here](https://openreview.net/forum?id=Pjky9XG8zP&noteId=vWAXDtIMwZ). It can be observed that **even with the budget aligned in different fashions, SR-GKP always dominates HARP on all tested benign and adv. metrics.** Please refer to the detailed results/discussion below.
> > > >
> > > >
> > > > ### **[Follow-up #2: Comparison with HARP]: Sure! SR-GKP dominates.**
> > > >
> > > > The only overlapped experiment between channel-pruned HARP (hereinafter "HARP-CP") and SR-GKP is VGG-16 , and here's the results:
> > > >
> > > > | Method | Baseline Budget | Pruning Budget | Fine-tuning Budget | Best Benign | Best FGSM $\varepsilon = 0.01$ | Best FGSM $\varepsilon = 0.1$ | Best PGD |
> > > > |:-:|:-:|:-:|:-:|:-:|:-:|:-:|:-:|
> > > > | HARP-CP | NT, 300 epochs (AS) | AT, 20 epochs (LS) | NT, 300 epochs (NS) | 91.66 | 55.11 | 23.13 | 3.30 |
> > > > | HARP-CP | NT, 300 epochs (NS) | AT, 20 epochs (LS) | NT, 300 epochs (NS) | 87.04 | 23.49 | 10.71 | 0.05 |
> > > > | HARP-CP | NT, 300 epochs (AS) | AT, 20 epochs (LS) | AT, 100 epochs (AS) | 72.06 | 64.08 | 17.96 | 50.93 |
> > > > | SR-GKP | NT, 300 epochs (LS) | 0 epoch | NT, 300 epochs (SS) | **93.95** | **83.85** | **59.03** | **66.62** |
> > > > | SR-GKP | NT, 300 epochs (LS) | 0 epoch | NT, 100 epochs (SS) | **93.48** | **82.62** | **52.35** | **62.18** |
> > > >
> > > > (AT = Adv. Training; NT = Natural Training; AS = epoch with best adv. acc saved; NS = epoch with best benign acc saved; LS = last epoch saved; SS = Superscore according to [Table 8](https://openreview.net/attachment?id=Pjky9XG8zP&name=supplementary_material#page=18).)
> > > >
> > > > **We believe it is safe to conclude that SR-GKP showcased clear dominates over the HARP-CP.**
> > > >
> > > > Although one network-dataset combination might not be a comprehensive evaluation (note, HARP-CP is only implemented on two models, as HARP is mainly proposed as a weight pruning method with a channel pruning extension offered in appendix), [Figure 13 of HARP](https://openreview.net/pdf?id=sAJDi9lD06L#page=21) suggests HARP-CP is extremely close to RAP-ADMM [4] — a method we already comprehensively compared — in terms of both benign and adv. acc; and from [Figure {6, 7, 8}](https://openreview.net/attachment?id=Pjky9XG8zP&name=supplementary_material#page=18) we may conclude that RAP-ADMM performs vastly below SR-GKP on various metrics. **We find it interesting as HARP and RAP-ADMM — two methods utilizing adv. training — still perform worse than SR-GKP on most adv. tasks, though SR-GKP only sees benign inputs; indicating the effectiveness of SR-GKP**. We suspect this is mainly due to HARP and RAP-ADMM are not structured pruning-focused methods.
> > > >
> > > > ### **[Follow-up #3: Comparison with MAD]: They are incomparable (unstructured vs structured).**
> > > >
> > > > MAD is an unstructured weight pruning method that prunes each weight individually — evident by the *"sparse"* notion in its title, its comparison with LWM and HYDRA (two weight pruning methods), and its 90% sparsity showcased in Table 1 — where our method lives in the structured pruning realm and prunes grouped kernels. They are incomparable due to reasons listed in [Section 1.1 - Structured vs Unstructured Pruning: Accuracy-Efficiency Trade-off](https://openreview.net/pdf?id=Pjky9XG8zP#page=2).
> > > >
> > > > ---
> > > >
> > > > **We hope the added info may convince the reviewer about the efficacy of our proposed method, and maybe warrant us a higher rating. Thanks again.**
> > > >
> > > > ---
> > > >
> > > > [1] Zhong et al., Revisit Kernel ... Convolutions. ICLR 2022
> > > > [2] Liu & Sun et el., Rethinking ... Network Pruning. ICLR 2019
> > > > [3] Li & Lin et al., Towards ... Collaborative Compression. CVPR 2021
> > > > [4] Ye et al., Adversarial Robustness ... Both? ICCV 2019

---

> > > > > ### Comment · Reviewer_xyz8 · 2023-08-19
> > > > > **Thanks to Response**
> > > > >
> > > > > The reviewer would like to express a gratitude to the authors re-pointing the training scheme experiment and explainig the definite difference compared with MAD. For leading readers who may be not in this area not to be confused, the reviewer highly recommends the authors to include **above  interesting discussion points with MAD, HARP**  by using the way the authors explained to the reviewer. Plus, the reviewer will improve the score rating.

---

> > > > > > ### Author Response · Authors · 2023-08-19
> > > > > > **Thank you, we will certainly incorporate such discussion.**
> > > > > >
> > > > > > The reviewer made a good point — particularly on the inclusion of MAD-like methods — and we will happily oblige. We the authors have dedicated a paragraph to talk about existing structured pruning methods that claim to provide adversarial improvements ([Line 136 -146](https://openreview.net/pdf?id=Pjky9XG8zP#page=4)), but failed to comprehensively list out works with adversarial benefits from the unstructured realm. **Though incomparable, we believe this is a noteworthy discussion** — in fact, the lack thereof might have been the very reason we didn't include a comparison with HARP in the first place; we, again, thank reviewers `sejQ` and `xyz8` for suggesting such an important comparison — **as scholars might be able to collect insights from unstructured methods to facilitate development in the structured side** (as it already happened countless time in the benign space, e.g., [1] to [2] to [3]). We will provide this brief rundown under the current [Section 2 - Related Work and Discussion](https://openreview.net/pdf?id=Pjky9XG8zP), as well as a more elaborated discussion in our current [Appendix B - Extended Related Work and Discussion](https://openreview.net/attachment?id=Pjky9XG8zP&name=supplementary_material#page=14).
> > > > > >
> > > > > > It is not often to receive a boost from `3` to `7`. **We thank you for keeping such an open-mindedness for our work and facilitating an engaging author-reviewer discussion.**
> > > > > >
> > > > > > ---
> > > > > > [1] Frankle & Carbin, The Lottery Ticket Hypothesis: Finding Sparse, Trainable Neural Networks. ICLR 2019
> > > > > > [2] Renda et al., Comparing Rewinding and Fine-tuning in Neural Network Pruning. ICLR 2020
> > > > > > [3] Zhong et al., Revisit Kernel Pruning with Lottery Regulated Grouped Convolutions. ICLR 2022

---

### Official Review · Reviewer_sejQ · 2023-07-05

**Soundness:** 2 fair
**Presentation:** 3 good
**Contribution:** 2 fair
**Rating:** 6
**Confidence:** 4

**Summary:**

This paper proposes a structured pruning method that introduces kernel smoothness into grouped kernel pruning to obtain adversarial robustness. Without additional overhead (compared to standard pruning methods), the method can obtain considerable accuracy and adversarial robustness simultaneously. The authors compare with popular structured pruning methods, and the results validates the effectiveness of the proposed method.

**Strengths:**

- The motivation is reasonable: introducing kernel metrics related to adversarial robustness into the pruning process to obtain robustness.

- The authors designed sufficient ablation experiments to validate that the proposed 'Smoothness Snaking' Filter Grouping strategy and 'Smooth Beam Greedy' pruning strategy are more effective than other strategies.

**Weaknesses:**

- The novelty is limited: kernel smoothness and grouped kernel pruning have been studied in some works. The core contribution of the paper seems to be a direct combination of two ideas.

- Some baselines are omitted in the experiments, such as HARP, which can be extended to structured pruning for robustness.

  [1] Holistic adversarially robust pruning.

- Lack of performance comparison under varying pruning ratio.

**Questions:**

- The author claims the method is one-shot. However, the one-shot pruning typically refers to the pruning method without the need of fine-tuning, such as OTO [2]. The proposed method seems still requiring fine-tuning afterwards.

  [2] Only train once: a one-shot neural network training and pruning framework.

**Limitations:**

Yes.

---

> ### Author Rebuttal · Authors · 2023-08-10
>
> We thank the reviewer for your time, as well as for recognizing the motivation of our task and the solidness of our approach. To address your mentioned weaknesses:
>
> ### **[W1 - Limited novelty]: Please refer to our [general comment](https://openreview.net/forum?id=Pjky9XG8zP&noteId=1EzOrtnZm3).**
>
> ### **[W2 - Comparison with HARP]: Sure! SR-GKP still dominates.**
>
> Good suggestion. We missed it as HARP is presented as a weight pruning paper with a channel pruning extension. The only overlapped experiment between channel-pruned HARP (hereinafter "HARP-CP") and SR-GKP is VGG-16, and here's the results:
>
> | Method | Baseline Budget | Pruning Budget | Fine-tuning Budget | Best Benign | Best FGSM w/ $\varepsilon = 0.01$ | Best FGSM w/ $\varepsilon = 0.1$ | Best PGD |
> |:-:|:-:|:-:|:-:|:-:|:-:|:-:|:-:|
> | HARP-CP | NT, 300 epochs (AS) | AT, 20 epoch (LS) | NT, 300 epochs (NS) | 91.66 | 55.11 | 23.13 | 3.30 |
> | HARP-CP | NT, 300 epochs (NS) | AT, 20 epoch (LS) | NT, 300 epochs (NS) | 87.04 | 23.49 | 10.71 | 0.05 |
> | HARP-CP | NT, 300 epochs (AS) | AT, 20 epoch (LS) | AT, 100 epochs (AS) | 72.06 | 64.08 | 17.96 | 50.93 |
> | SR-GKP | NT, 300 epochs (LS) | 0 epoch | NT, 300 epochs (SS) | 93.95 | 83.85 | 59.03 | 66.62 |
> | SR-GKP | NT, 300 epochs (LS) | 0 epoch | NT, 100 epochs (SS) | 93.48 | 82.62 | 52.35 | 62.18 |
>
> (AT = Adversarial Training; NT = Natural Training; AS = epoch with best adv. acc is saved; NS = epoch with best benign acc is saved; LS = last epoch is saved; SS = Superscore according to [Table 8](https://openreview.net/attachment?id=Pjky9XG8zP&name=supplementary_material#page=18).)
>
> **We believe it is safe to conclude that SR-GKP showcased clear dominates over the HARP-CP.**
>
> Although one network-dataset combination is not a comprehensive evaluation, Figure 13 of HARP ([OpenReview link](https://openreview.net/pdf?id=sAJDi9lD06L#page=21)) suggests HARP-CP is extremely close to RAP-ADMM [1] — a method we already compared — in terms of both benign and adversarial acc; and from [Figure {6, 7, 8}](https://openreview.net/attachment?id=Pjky9XG8zP&name=supplementary_material#page=18) we may conclude that RAP-ADMM performs vastly below SR-GKP on various metrics. **We find it interesting as HARP and RAP-ADMM — two methods utilizing adversarial training — still perform worse than SR-GKP on most adversarial tasks, though SR-GKP only sees benign inputs; indicating the effectiveness of SR-GKP**. We suspect this is mainly due to HARP and RAP-ADMM are not structured pruning-focused methods.
>
> We will also include a discussion about HARP in our related work session; again, we thank the reviewer for mentioning this work.
>
> ### **[W3 - More pruning ratios]: Sure! Again, SR-GKP is going strong.**
>
> Here's $PR \approx 62.5\%$. Given our raw experiment results would be massive like [Table {9, 10, 11}](https://openreview.net/attachment?id=Pjky9XG8zP&name=supplementary_material#page=20), we plotted a couple graphs to facilitate your reading — please kindly direct your attention to the to-all-reviewer comment, as that's where we may attach a PDF ([redirection to general comment](https://openreview.net/attachment?id=1EzOrtnZm3&name=pdf)).
>
> We believe it would be fair to say that SR-GKP still delivers the best comprehensive performance with $PR \approx 62.5\%$. Methods like SFP and FPGM still stay as the best competitors, but they generally cannot deliver consistently good benign & adversarial performances across different depths of ResNets, and are weak on ImageNet benign readings. SR-GKP seems to be the only constantly strong method among all tested experiments and metrics (within top #3 for almost all trials).
>
> ---
> ### **[Q1 - Ambiguity on term "one-shot."]: Our definition is probably more main-stream.**
>
> This particular term, much like "kernel," carries different connotations under the pruning context. We are confident that our utilization of one-shot — which implies the pruning procedure is done all at once with no training/fine-tuning done in between (in contrast to iterative pruning, where a prune-train cycle is required) — is a popular, if not the more "mainstream," definition; this is evident by the following heavily-cited pruning papers [2, 3, 4]:
>
> > [3]: *"... this pruning approach is one-shot: the network is trained once, p% of weights are pruned, and the surviving weights are reset (then retrained)"*
> [4]: *"... the outline above prunes the network to a target sparsity level all at once, known as one-shot pruning."*
>
>
> We authors are also familiar with the OTO procedure, and it is our understanding that OTO is more like a *from-scratch* (in contrast to post-train) *iterative pruning* method, as it iteratively introduces group sparsity during the training of a (often time, randomly initialized) model. Section 3 of [5] ([OpenReview Link](https://openreview.net/pdf?id=7ynoX1ojPMt#page=3)) and hyperparameters like `start_pruning_steps` ([OTO GitHub Link](https://github.com/tianyic/only_train_once/blob/main/tutorials/01.quick_start_resnet18_cifar10.ipynb)) support our understanding.
>
> We plan to stay on using this "one-shot" to describe SR-GKP, though we understand there are popular papers, e.g., SparseGPT [6], that interpret one-shot pruning as the reviewer. We will add a proper clarification for disambiguation.
>
> ---
> Ref.
> [1] Ye et al., Adversarial Robustness vs Model Compression, or Both? ICCV 2019
> [2] Liu & Sun et el., Rethinking the Value of Network Pruning. ICLR 2019
> [3] Frankle & Carbin, The Lottery Ticket Hypothesis: Finding Sparse, Trainable Neural Networks. ICLR 2019 (Best Paper Award)
> [4] Renda et al., Comparing Rewinding and Fine-tuning in Neural Network Pruning. ICLR 2020
> [5] Chen et al., OTOv2: Automatic, Generic, User-Friendly. ICLR 2023
> [6] Frantar and Alistarh, SparseGPT: Massive Language Models Can be Accurately Pruned in One-Shot. ICML 2023

---

> > ### Author Response · Authors · 2023-08-21
> > **Summary of our rebuttal, as well as an invitation to discuss.**
> >
> > Currently, with `7765`, all reviewers' ratings are on the positive side of the scale (thanks!). Though — despite your initial positive rating of `5` — you are the only reviewer yet to engage our rebuttal. We understand that the reviewing pressure is heavy, and we all have personal matters to attend to, but please excuse us for urging as we want to ensure we have adequately addressed your concerns. Here, we provide a contextual summary.
> >
> > Just like `rEnW`, we believe it is fair to say **your initial feedback is plenty positive: as you recognize our work to score the trifecta of *"motivation being reasonable, method being solid, and evaluation to be able to validate our method's effectiveness."*** We, again, appreciate your recognition.
> >
> > Your concerns can be described as a tri-fold: 1) additional experiment requests on HARP and more pruning ratio; 2) use of the term "one-shot" being ambiguous; and 3) limited novelty due to both GKP and kernel smoothness have been studied before.
> >
> > We believe **we have addressed your first concern in a point-blank fashion, delivering results as requested, with both added experiments showcasing the superiority of our method.** For your second cosmetic/presentation concern, we justified our choice of the term by quoting popular pruning arts; meanwhile, we recognize the root of your concerns (as it is indeed ambiguous), and will clarify it should there be a camera-ready version.
> >
> > ---
> >
> > Regarding your third concern (novelty), again, we would like to kindly direct your attention to our [General Comment](https://openreview.net/forum?id=Pjky9XG8zP&noteId=1EzOrtnZm3). We agree that our method looks like a direct combination of GKP + kernel smoothness *from a glance*. However, we respectfully note **this combination is unique to GKP** (as [it does not work](https://openreview.net/attachment?id=Pjky9XG8zP&name=supplementary_material#page=16) on traditional structured granularity, e.g., filters), **so it takes insight and field ownership to spot the opportunity**. Yet, **such mix-up is non-trivial to figure out**, and the showcased design is a result of a series of careful investigatory studies ([C.1.1-C.1.2](https://openreview.net/attachment?id=Pjky9XG8zP&name=supplementary_material#page=14)).
> >
> > We argue that **principally landing on a solution with a simple design should not discount its contribution**, especially when having a less sophisticated method helps us better deliver our paper's main message (GKP variants being superior on benign and adversarial tasks) while fitting in the page limitation already populated by the heavy but necessary background intros (GKP, structured pruning, adversarial robustness, kernel metrics, etc.).
> >
> >
> > (We also further clarify — given the reviewer specifically notes *"grouped kernel pruning has been studied in some works"* — that **GKP is a pruning granularity, much like filter pruning, but not a specific method.** Thus, the fact that GKP has been proposed before should not discount further study under its realm — much like [1] being a follow-up GKP method to [2], and the countless filter pruning papers after its initial proposal [3, 4].
> >
> > That being said, we understand the reviewer might not mean it literally, but was rather commenting on the framework resemblance between our method and [2]. If this is the case, we like to emphasize that our work is the first to comprehensively study adversarial robustness under the structured pruning context; it just so happened that we leveraged GKP's power of being a finer densely structured granularity. On the technical aspect, we hope our [comment](https://openreview.net/forum?id=Pjky9XG8zP&noteId=5JZwTCI79J) on SR-GKP's execution efficiency and performance gap may provide you with more relevant info, as it sure [convinced `xyz8`](https://openreview.net/forum?id=Pjky9XG8zP&noteId=pbmKddnl5q); who previously have the same novelty concern as you do.)
> >
> > ---
> > [1] Park et al., Dynamic Structure Pruning for Compressing CNNs. AAAI 2023 (Oral)
> > [2] Zhong et al., Revisit Kernel Pruning with Lottery Regulated Grouped Convolutions. ICLR 2022
> > [3] Li et al., Pruning Filters for Efficient ConvNets. ICLR 2017
> > [4] Zhou et al., Less is More: Towards Compact CNNs. ECCV 2016

---

> > > ### Author Response · Authors · 2023-08-21
> > > **Rank-based presentation to help you digest the mass results.**
> > >
> > > We also understand our [plots](https://openreview.net/attachment?id=1EzOrtnZm3&name=pdf) for the requested alternative pruning ratio experiments can be hard to digest due to the extended y-axis range and the large number of methods/metrics involved. Here, we provide a ranking chart to improve its readability.
> > >
> > >
> > > > **Table of methods ranked against each other on each model with pruning rate $\approx 62.5\%$ , lower is better.** *"ResNet-XX Mean Rank"* means a method's average ranks across four metrics on ResNet-XX; e.g., SR-GKP is ranked `#1/#1/#3/#3` for its best performance across `benign/FGSM 0.01/FGSM 0.01/PGD` metrics on ResNet-110 against other methods, so it'd have a *ResNet-110 Mean Rank* of `(1+1+3+3)/4 = #2`. Methods with incomplete presence across the three models are excluded (namely, HRank and LRF).
> > >
> > > | Method | ResNet-32 Mean Rank | ResNet-56 Mean Rank | ResNet-110 Mean Rank | All Models Mean Rank |
> > > |---|:-:|:-:|:-:|:-:|
> > > | CC | **#1** | #3.5 | #5.75 | #3.42 |
> > > | DHP | #3.75 | #7.5 | #7.75 | #6.33 |
> > > | FPGM | #4.25 | #3.75 | **#2** | #3.33 |
> > > | L1Norm-A | #5.5 | #3.75 | #5.75 | #5.00 |
> > > | L1Norm-B | #6.25 | #6.25 | #4.25 | #5.58 |
> > > | NPPM | #5 | #4 | #5.5 | #4.83 |
> > > | SFP | #7.5 | #5 | #3 | #5.17 |
> > > | SR-GKP (Ours) | #2.75 *(2nd-best)* | **#2.25** | **#2** | **#2.33** |
> > >
> > >
> > > **We believe the above table clearly demonstrated our methods' dominance at an alternative/more aggressive pruning ratio.** (If you find such table format helpful, please also let us know so we will provide such tables for all our mass experiments, should there be a camera-ready version).
> > >
> > > Again, given we have added most experiments per your requests, **we would love to receive your feedback and hopefully confirm our added experiments/discussion have addressed your concerns.**

---

> > > > ### Comment · Reviewer_sejQ · 2023-08-21
> > > >
> > > > Thanks for the impressive rebuttal. I am satisfied with most of your comments. Yet the interpretation of **main trend** of "one-shot" prevents me from increasing my rating. As the author have applied OTO library onto many experiments, I think that you might have realized the ease-of-use of the OTO library compared to the previous "one-shot" works. Considering from the view of the end-users who are not expertsing on pruning, which one-shot they might prefer more? Therefore, I would recommend that in the revision, not argue which branch is main-trend, yet describe them equally.

---

> > > > > ### Author Response · Authors · 2023-08-21
> > > > > **Agree that OTO scores much higher on the usability scale. We will sure clarify the difference between our "one-shot" vs the OTO one.**
> > > > >
> > > > > The reviewer made a good point that the "mainstream-ness" of a definition might not be a bulletproof defense when coming to term definition, though, popularity is a compromise we have to live by with term-overlap issues (as an anecdote, the OTO submission to NeurIPS 21 also [faced scrutiny](https://openreview.net/forum?id=p5rMPjrcCZq&noteId=6sVPeNFqCpk) on its use of "one-shot", where the reviewer adopted the definition we are using; the [author response](https://openreview.net/forum?id=p5rMPjrcCZq&noteId=IVdQGSRZsX) also confirmed it is actually on a multi-stage procedure). We'd agree that outside the pruning field, the definition of "one-shot" is more in-line with the OTO one.
> > > > >
> > > > > And yes, **we authors are impressed by the OTO library for its easy-of-use and being architecture-agnostic**, and the usability of our method is, frankly, vastly behind OTO (for a start, it only works on CNNs). For disambiguation, we will introduce our method to be a **"one-shot structured pruning method for CNN that follows the classic *train - prune - fine-tune* procedure, where all excessive components are pruned all at once before fine-tuning." We would also specifically make a footnote to distinguish it from OTO's "one-shot" regarding usability/architecture support to avoid any potential confusion.**
> > > > >
> > > > > Would this adjustment resolve the reviewer's concern and maybe warrant us a higher rating?

---

> > > > > > ### Comment · Reviewer_sejQ · 2023-08-21
> > > > > >
> > > > > > Thank you for the clarification that resolve my concerns. Therefore, I decided to increase my rating to 6.

---

### Official Review · Reviewer_Tjyo · 2023-07-07

**Soundness:** 3 good
**Presentation:** 2 fair
**Contribution:** 3 good
**Rating:** 7
**Confidence:** 3

**Summary:**

The contribution of the paper is two fold. First, it presents a large scale benchmarking study of the adversarial robustness of various filter pruning methods. It finds that such methods often lead to a large degradation in the adversarial robustness of the pruned models, though they preserve benign accuracy. The paper then proposes an alternative form of pruning to solve this issue. The proposed method borrows ideas from grouped kernel pruning and kernel smoothness to prune out groups of kernels while making sure that the model does not overfit to adversarially vulnerable high frequency components of the image. This leads to better adversarial performance than baselines.

**Strengths:**

1. The empirical evaluation of the paper is solid.
2. The paper proposes a simple hypothesis of pruned models fitting to HFCs to explain their poor robustness, and unit-tests it well.
3. The proposed method effectively combines two existing works, and provides solid gains.

**Weaknesses:**

1. The writing of the paper can be improved. There are multiple typos, and the paper could benefit from formally defining terms and notations earlier on. A more thorough discussion of the GKP and kernel smoothing techniques would also help.
2. The claim of no additional training cost over standard pruning methods is not verified.

**Questions:**

1. Can the authors provide information about the training cost

**Limitations:**

The limitations are sufficiently discussed.

---

> ### Author Rebuttal · Authors · 2023-08-10
>
> We thank reviewer `Tiyo` for your appreciation of our work, especially in terms of the benchmark and ablation study parts. We invested a lot of time/manpower/compute into them, as we believe such results and to-be-open-sourced codebase may greatly help out the structural pruning community to efficiently identify problems and conduct fair comparison. Yet we hope our ablation studies may facilitate future development in the GKP realm.
>
> ### **[W1 - Term clarification and typos]: Thanks!**
> We will address those ambiguities/typos, as well as give the paper a detailed polish, should there be camera-ready version.
>
> ### **[W2, Q1 - Show support for SR-GKP not requiring additional training cost]: Sure!**
>
>
> Under the classic *train - prune - fine-tune/retrain* paradigm (where our method lives), the training cost is determined at two stages: training the unpruned baseline model and fine-tuning the pruned model.
>
> We claim SR-GKP carries no additional training cost as it is a post-train pruning method, so it won't affect the training cost of the first stage. As for fine-tuning, the SR-GKP one-shot pruned model is already compressed and will go through a vanilla SGD update with no extra operation; we also specifically ensured that all compared methods are using an identical (or similar to the best of the method's ability) epoch budget for fine-tuning — thus, it requires no additional (in fact, often less) cost on the fine-tuning stage as well.
>
> For illustration convenience, here we walk through two exemplary comparisons:
>
> **Compares to TMI-GKP [1]** — the founding method of Grouped Kernel Pruning — a SR-GKP pruned model shares the exact same format with a TMI-GKP pruned mode, then such pruned models will go through the identical fine-tuning procedure. **In this case, SR-GKP yields no additional training/fine-tuning cost.** Additionally, we'd like to direct your attention to Table 7 ([page 16](https://openreview.net/attachment?id=Pjky9XG8zP&name=supplementary_material#page=16)), which indicates SR-GKP has a much faster execution time than TMI-GKP in terms of pruning procedure, thus being a more "executionally efficient" method overall.
>
> **Compare to SFP [2]** — a popular filter pruning method exhibits strong adversarial performance — where its pruned model is zero-masked. Such zeroed weights will reactivate during its retraining/fine-tuning, where the algorithm decides which filter to zero out between epochs. **In this case, the one-shot pruned SR-GKP induces significantly less cost** as a) it does not require weight update to the full model, but only on a compressed one; and b) it does not perform extra pruning operation during the fine-tuning/retraining stage.
>
> We hope our response clarifies the reviewer's questions and further demonstrates the effectiveness of our method, and we will add similar information to D.1.3 ([page 17](https://openreview.net/attachment?id=Pjky9XG8zP&name=supplementary_material#page=17)) to enrich the clarity of our method comparison. Given the borderline scoring of our work, we hope the reviewer may consider raising the score, especially on the presentation aspect.
>
> ---
>
> Ref.
> [1] Zhong et al., Revisit Kernel Pruning with Lottery Regulated Grouped Convolutions. ICLR 2022
> [2] Ye et al., Soft Filter Pruning for Accelerating Deep Convolutional Neural Networks. IJCAI 2018

---

> > ### Comment · Reviewer_Tjyo · 2023-08-13
> > **Response**
> >
> > I thank the authors for their rebuttal, and maintain my score recommending acceptance.

---

### Author Rebuttal · Authors · 2023-08-10

# General Response

We thank all reviewers for your valuable time and comments. We are glad that many reviewers found:

* **Our task to be well-motivated**
    * `rEnW`: *"The problem considered is important."*
    * `sejQ`: *"The motivation is reasonable."*
* **Our approach to be technically sound**
    * `rEnW`: *"The method is clever and technically sound."*
    * `xyz8`: *"The paper is technically sound."*
    * `Tjyo`: *"The paper proposes ... and unit-tests it well."*
    * `Tjyo`: *"The proposed method effectively combines two existing works, and provides solid gains."*
    * `sejQ`: *"The authors designed sufficient ablation experiments to validate that the proposed ... strategies."*
* **Our empirical evaluation to be solid**
    * `Tjyo`: *"The empirical evaluation of the paper is solid."*
    * `rEnW`: *"Empirical evidence shows merit in using this approach."*

Again, we are grateful for your recognition.

In terms of limitations, other than the cosmetic issues and some factual remarks, two common notions are:

* **More experiments**
    * `sejQ` asked for the inclusion of HARP [1], a robustness-aware weight pruning method with a channel pruning extension. We delivered.
    * `sejQ` asked for more variety in pruning rates. We delivered.
    * `rEnW` asked for more ImageNet baseline. We explained how we reached such baseline selection and delivered the asked comparison to the best of their repo's reproducibility; we even went beyond and delivered the reproducible ones on CifarResNets.
* **Limited novelty**
    * `sejQ`: *"The novelty is limited: kernel smoothness and grouped kernel pruning have been studied in some works. The core contribution of the paper seems to be a direct combination of two ideas."*
    * `xyz8`: *"The novelty of the method is not high... Even if the paper has a method with smoothness, it still sounds like one of the variants of TMI-GKP under the same umbrella."*

While novelty is certainly important, we believe the novelty of a paper can be roughly viewed from two aspects: **empirical novelty** — as if the paper unveils properties and behavior that is unknown to the general audience (e.g., lottery ticket style [2]); and **technical novelty** — as if the propose solution designs anything new to achieve its goal (e.g., RWKV [3]).

**We argue that our work is rich in empirical novelty, as before it, one cannot answer the following important questions with certainty**:
1. Are carefully designed structured pruning methods with similar benign performance on par with each other on adversarial tasks? Are newer methods more robust?
2. Is Grouped Kernel Pruning naturally more robust to adversarial attacks than filter/channel pruning?
3. Is adversarial training absolutely necessary for obtaining a robust pruned model?
4. Is it possible to have a structurally pruned model that performs well under both benign and adversarial tasks?

In terms of technical novelty, we agree that our method looks like a direct combination of kernel smoothness and GKP from the surface. However, we'd like to emphasize that the **combination between kernel smoothness and GKP is unique**, as mixing kernel smoothness  — a kernel-level metric — with filter pruning scheme will result in a catastrophic performance drop ([C.1.3](https://openreview.net/attachment?id=Pjky9XG8zP&name=supplementary_material#page=16)). Yet, **such mix-up is non-trivial to figure out**, as a direct mixture will not work, so we have to apply principled ablation studies to find out the right way of integration ([C.1.1-C.1.2](https://openreview.net/attachment?id=Pjky9XG8zP&name=supplementary_material#page=14)).


**We also intensionally kept our method simple**, as GKP is an extremely young pruning family with only three well-developed methods under its wing [4, 5, 6]. Though it has gained some recognition for its strong benign performance (with [6] winning AAAI Oral), the procedure and capability of GKP can be foreign to many readers. Under this context, if we proposed a very sophisticated and trick-heavy GKP method that performs well on adversarial tasks, one would likely question if such gains are from the extra addition, or is it due to GKP granularity being naturally more adversarially robust and more flexible on incorporating robustness-related maneuvers?

We argue that with the necessity of including backgrounds on GKP, structured pruning, adversarial robustness, (tools like) kernel smoothness, and a vast amount of empirical results, **a relatively vanilla GKP implementation is the most appropriate vehicle to deliver our claim: that GKP variants are superior on both benign and adversarial tasks**, and thus, *"one less reason for filter pruning."*

We further note — though we believe our reviewers and AC are well-aware — that technical novelty is not the only important contribution. We believe our investigation on how to do GKP differently ([Section 3.2.1](https://openreview.net/attachment?id=Pjky9XG8zP&name=supplementary_material#page=6)), benchmark report on various methods against different evaluation metrics, as well as our to-be-open-sourced codebase and checkpoints for fair and comprehensive evaluation... **may provide essential benefits to future GKP developers and the pruning community at large.**


---
Ref.
[1] Zhao & Wressnegger, Holistic Adversarially Robust Pruning, ICLR 2023
[2] Frankle & Carbin, The Lottery Ticket Hypothesis: Finding Sparse, Trainable Neural Networks. ICLR 2019
[3] Peng et al., RWKV: Reinventing RNNs for the Transformer Era. arXiv 2023
[4] Zhong et al., Revisit Kernel Pruning with Lottery Regulated Grouped Convolutions. ICLR 2022
[5] Zhang et al., Group-based network pruning via nonlinear relationship between convolution filters. Applied Intelligence, 2022
[6] Park et al., Dynamic Structure Pruning for Compressing CNNs. AAAI 2023

---

---

### Author Response · Authors · 2023-08-21
**To AC (and Thank You to All Reviewers).**

Dear Area Chair,

We are blessed to receive an engaging author-reviewer discussion and glad that all reviewers are on the positive side of the scale, with many vastly improved their scores per our rebuttals (from `7543` to `7766`).

Again, as noted in our *General Response* below, we are grateful that most reviewers found our work to score the trifecta of **task being well-motivated (`rEnW` and `sejQ`),  method being technically sound (`rEnW`, `xyz8`, `Tjyo`, `sejQ`), and empirical evaluation to be solid (`rEnW`, `Tjyo`, `sejQ`)** — we cannot ask for more of this type of work.

In terms of limitations — other than some factual remarks and cosmetic issues — `sejQ`, `rEnW`, and `xyz8` asked for **additional results** on more baselines and an alternative pruning ratio, and we delivered such requests. All added results are supportive of our approach, and all three reviewers are satisfied with our responses (Response of [`sejQ`](https://openreview.net/forum?id=Pjky9XG8zP&noteId=ZppD2j1goT), [`rEnW`](https://openreview.net/forum?id=Pjky9XG8zP&noteId=oT3729h2EV), and [`xyz8`](https://openreview.net/forum?id=Pjky9XG8zP&noteId=OUOXzy5MtZ)). `sejQ` and `xyz8` also initially raised concerns about the **novelty** of our work, but both were convinced after reading our General Response and per-reviewer discussions ([w/ `sejQ`](https://openreview.net/forum?id=Pjky9XG8zP&noteId=0VvfsvV5ci) and [w/ `xyz8`](https://openreview.net/forum?id=Pjky9XG8zP&noteId=5JZwTCI79J)); with [`sejQ` commented](https://openreview.net/forum?id=Pjky9XG8zP&noteId=ZppD2j1goT) our rebuttal to be *"impressive"* and [`xyz8` *"respectfully agree*](https://openreview.net/forum?id=Pjky9XG8zP&noteId=pbmKddnl5q) *with the author’s opinion on the novelty of the paper"* then notably improved the rating from `3` to `7` (THANKS!). Then, there are some **term ambiguity issues** raised in a general (`Tjyo`) or specific manners ([`sejQ`](https://openreview.net/forum?id=Pjky9XG8zP&noteId=ZppD2j1goT) and `rEnW`), we believe these are well-addressed after our thorough discussion with the specific reviewers, and we will sure give the paper a detailed polish should there be a camera-ready. Last, `xyz8` commented on the lack of **theoretical analysis** of our work, which we agree, but we also [respectfully note](https://openreview.net/forum?id=Pjky9XG8zP&noteId=Ps65DaKw2J) that this is very much the norm of the structured pruning field, and showcased why a provable justification might be beyond the currently available theoretical instruments.

We also collected many useful insights and suggestions on what to include/adjust (e.g., discussions regarding [schools of implementation choices](https://openreview.net/forum?id=Pjky9XG8zP&noteId=dLmmsgOfzV) and related work intros from the [unstructured realm](https://openreview.net/forum?id=Pjky9XG8zP&noteId=OUOXzy5MtZ)), which we will happily add to our updated paper — **we sincerely appreciate the thoughtful suggestions by the reviewers.**

We believe our current rating might be on the edge of being award-worthy. While we would, of course, wholeheartedly respect your final decision; **if we may, we like to boldly advocate that an award for this work might be beneficial to the pruning and adv. communities at large.** Since we are the first to comprehensively study the field of adv. robust structured pruning, we will provide **a toolkit to *replicate* and benchmark** various pruning methods against different evaluation metrics, and we believe our to-be-open-sourced **codebase and checkpoints may facilitate a fair and comprehensive evaluation for all future pruning works** (regardless in the adv. space or not). **Many reviewers also share the same view** (`Tjyo` on the note of our benchmark, [`sejQ`](https://openreview.net/forum?id=Pjky9XG8zP&noteId=ZppD2j1goT) & [`xyz8`](https://openreview.net/forum?id=Pjky9XG8zP&noteId=OUOXzy5MtZ) by providing suggestions to facilitate audience beyond the pruning field, [`rEnW`](https://openreview.net/forum?id=Pjky9XG8zP&noteId=bNm0JwtGYk) for recommending content inclusion to benefit the *"community at large."*).

We also believe our work may adequately utilize the award exposure, as outside than proposing a sound pruning method, our main intent is to direct attention to the **important but often overlooked** field of adv. robust structured pruning — an area gained high traffic in the unstructured realm, while being arguably more relevant under the structured context ([Line 76-80](https://openreview.net/pdf?id=Pjky9XG8zP#page=3)) — and to attract more scholars to ***Grouped Kernel Pruning***, a granularity with many natural benefits over the traditional filter pruning, thus worthy of further exploration. Please [refer to **Line 92-118**](https://openreview.net/pdf?id=Pjky9XG8zP#page=3) for our detailed claims, contributions, and **community offers.**

Again, we thank you for your time and efforts in organizing and overseeing the paper review process.

Sincerely,
*Paper 2351* Authors

---

### Decision · Program_Chairs · 2023-09-21

**Decision:**

Accept (poster)

**Comment:**

Although the reviewers acknowledged the importance of the problem tackled, the motivation behind this work, the empirical evaluation, and the soundness of the method, they expressed some concerns about the writing of the paper, the verification of some claims, the novelty of the proposed method, the lack of comparison to some baselines, the ambiguity of some terms ("one-shot", "data-free"), and the lack of theoretical analysis. The authors' feedback and the thorough discussion between the reviewers and the authors eventually convinced all reviewers, who reached a consensus for acceptance.

The AC agrees that this study, at the intersection of network pruning and adversarial robustness, is of interest to a large portion of the community. The results are interesting, and the method, although relatively simple, is intuitive and effective. As such, this paper can be accepted to NeurIPS. However, the AC also agrees with the reviewers that writing quality should be improved and urges the authors to carefully review their paper for the final version.